# PARP10 is critical for stress granule initiation

Aravinth Kumar Jayabalan[1], Krishna Bhambhani[1,2], Anthony KL Leung[1,3,4,5]

**Stress granules (SGs) are cytoplasmic biomolecular condensates enriched with RNA and translation factors. They form in response to stress, in part through phosphorylation of the translation initiation factor eIF2α, and are implicated in viral infection, tumorigenesis, and neurodegeneration. Although ADP-ribosylation plays a key role in SG assembly, the enzyme responsible for this ADP-ribosylation during SG assembly remains unidentified. Here, we systematically knock down the human ADP-ribosyltransferase family and identify PARP10 as pivotal for SG assembly. Live-cell imaging reveals PARP10's crucial role in regulating initial SG assembly kinetics. Further, we pinpoint the core SG component, G3BP1, as a PARP10 substrate and find that PARP10 regulates SG assembly via G3BP1 or a synthetic mimic that recapitulates its domain architecture. PARP10 knockdown reduces eIF2α phosphorylation and alters the SG core composition, notably decreasing translation factor presence. Based on our findings, we propose a model in which ADP-ribosylation acts as a rate-limiting step, initiating the formation of SGs.**

## Introduction

Stress granules (SGs) are cytoplasmic condensates formed in response to various stressors, including viral infection and oxidative stress (Riggs et al, 2020; Glauninger et al, 2022). The assembly and disassembly of SGs are tightly regulated processes, occurring in response to stress and during relief from stress, respectively (Wheeler et al, 2016). Implicated in tumorigenesis, neurodegeneration, and antiviral innate immunity, elucidating the mechanisms underlying SG formation could offer new therapeutic avenues for various diseases (Grabocka & Bar-Sagi, 2016; McCormick & Khaperskyy, 2017; Advani & Ivanov, 2020; Mitrea et al, 2022; Jayabalan et al, 2023; Kassouf et al, 2023).

Upon stress, cells halt global translation—a process commonly mediated by the phosphorylation of the translation factor eIF2α—resulting in the release of mRNAs from ribosomes (Riggs et al, 2020). These mRNAs engage in RNA-RNA interactions, condensing with one another, as well as with translation factors and RNA-binding proteins in the cytoplasm to form SGs (Riggs et al, 2020; Ripin & Parker, 2023). Post-translational modifications further fine-tune this process, providing precise spatiotemporal control over SG assembly and disassembly (Tourrière et al, 2003; Ohn et al, 2008; Leung et al, 2011; Jayabalan et al, 2016; Marmor-Kollet et al, 2020; Gwon et al, 2021). The exact mechanisms that initiate SG assembly remain to be defined.

ADP-ribosylation—the addition of one or more ADP-ribose units onto proteins—plays roles in diverse biological processes (Gupte et al, 2017), such as DNA damage repair, mRNA translation, protein degradation, and the formation of biomolecular condensates, including SGs (Leung et al, 2011; Leung, 2020). Of the 17 ADP-ribosyltransferases in humans, commonly known as PARPs, all except PARP9 and PARP13 are catalytically active: PARPs 1, 2, 5a, and 5b add multiple ADP-ribose units, forming poly(ADP-ribose) or PAR, while the remaining 11 PARPs add single units (Gupte et al, 2017; Lüscher et al, 2022), resulting in mono-ADP-ribosylation or MARylation (Lüscher et al, 2022).

Five PARPs (PARPs 5a, 12, 13, 14, 15) and PAR glycohydrolase (PARG) are known SG components through immunostaining analyses (Leung et al, 2011; Catara et al, 2017; Duan et al, 2019). Proteomics analyses have further confirmed PARP12 and PARG as components in biochemically purified SGs (Jain et al, 2016; Youn et al, 2018). Overexpressing PARG suppresses SG formation, whereas the knockdown of this PAR-degrading enzyme impedes their disassembly (Leung et al, 2011; Duan et al, 2019). Therefore, PAR is key for SG assembly, acting as a structural scaffold and facilitating protein recruitment to SGs (Catara et al, 2017; Leung, 2020; Dasovich et al, 2021; Jayabalan et al, 2021). Despite the importance of PAR in SG assembly, the role of MARylation remains elusive, even though three out of five PARPs localized in SGs add MAR only. Intriguingly, overexpression of any of these PARPs can induce SGs (Leung et al, 2011). Our recent work further revealed that a conserved MAR-degrading enzyme in the non-structural protein 3 (nsP3) of alphaviruses is critical for viral replication, pathogenesis, and SG disassembly. While infection initially triggers SG formation, subsequent disassembly relies on nsP3's MAR-degrading activity (Jayabalan et al, 2021).

[1]Department of Biochemistry and Molecular Biology, Bloomberg School of Public Health, Johns Hopkins University, Baltimore, MD, USA   [2]Department of Chemical and Biomolecular Engineering, Whiting School of Engineering, Johns Hopkins University, Baltimore, MD, USA   [3]Department of Genetic Medicine, School of Medicine, Johns Hopkins University, Baltimore, MD, USA   [4]Department of Oncology, School of Medicine, Johns Hopkins University, Baltimore, MD, USA   [5]Department of Molecular Biology and Genetics, School of Medicine, Johns Hopkins University, Baltimore, MD, USA

Correspondence: anthony.leung@jhu.edu

Upon stress, RNA-binding proteins, including G3BP1, undergo increased ADP-ribosylation (Leung et al, 2011; McGurk et al, 2018; Duan et al, 2019). G3BP1 and its paralog G3BP2 are central to SG assembly, and their absence renders cells unable to form SGs under various stresses (Kedersha et al, 2016; Guillén-Boixet et al, 2020; Yang et al, 2020). Apart from being ADP-ribosylated, G3BPs can bind to PAR (Jayabalan et al, 2021). Notably, during infection, the viral-encoded MAR-degrading nsP3 also removes ADP-ribosylation from G3BP1 (Jayabalan et al, 2021). These findings underscore the importance of ADP-ribosylation of G3BP1 in SG formation.

Despite recognizing ADP-ribosylation's crucial role in SG assembly, we still lack clarity on which PARP(s) regulate this process, their exact roles, and targets within SGs (Leung et al, 2011; Duan et al, 2019; Leung, 2020; Jayabalan et al, 2021). Addressing these gaps could pave the way for developing inhibitors to modulate SG formation therapeutically (Mitrea et al, 2022). Through systematic knockdown, we identified PARP10—which adds MAR only (Kleine et al, 2008; Vyas et al, 2014)—as a critical factor for SG assembly. Our live-cell imaging demonstrated that diminished PARP10 levels delay SG formation, and our fractionation experiment showed reduced translation factor levels in the SG core upon PARP10 knockdown. We also identified G3BP1 as a substrate of PARP10 and found that PARP10 knockdown reduces eIF2α phosphorylation. We conclude with a model proposing how MARylation regulates SG assembly kinetics, particularly during the initial stage.

## Results

### MARylation is critical for SG assembly

Previously, we observed a notable difference in suppressing SG assembly by expressing hydrolase that removes PAR or MAR (Jayabalan et al, 2021). Human PARG degrades PAR, but not the last ADP-ribose conjugated to the protein (Brochu et al, 1994), thereby effectively converting PAR to MAR, while chikungunya virus nsP3 removes MAR from the protein (Abraham et al, 2018). Compared with the PAR-degrading PARG, the MAR-degrading nsP3 has a stronger effect in suppressing SG assembly (Jayabalan et al, 2021). These findings led us to hypothesize that MARylation is critical for SG assembly.

To further investigate the effect of MAR versus PAR hydrolases on SG assembly, we co-expressed PARG and nsP3, varying the DNA construct concentration of one while keeping the other constant. First, we maintained the amount of plasmid encoding the PAR-degrading enzyme PARG constant while gradually increasing the one for MAR-degrading nsP3. Typically, treating U2OS cells with 0.2 mM arsenite for 30 min induces SG assembly in ~90% of cells. However, expressing PARG alone reduced the number of cells with SGs to ~60%, as evidenced by the increased diffused distribution of the marker eIF3b (Fig 1A and B, red bars). More critically, co-expressing nsP3 with PARG significantly enhanced this reduction, resulting in merely ~30% of cells exhibiting SGs (Fig 1A–C). Notably, even minimal co-expression of nsP3 had a pronounced effect on SG inhibition.

In contrast, the expression of nsP3 alone results in SGs in only about 30% of cells, demonstrating a substantial inhibitory effect. The additional co-expression of PARG caused only minor further inhibition (Fig 1A and B, green bars). One potential explanation for this reduced amount of SGs is that the expression of these hydrolases may decrease arsenite-induced translational arrest, which would result in reduced phosphorylation of the translation initiation factor eIF2α (p-eIF2α). However, analysis of p-eIF2α levels under these conditions revealed no significant changes, suggesting that neither nsP3 nor PARG affects stress-induced translational arrest (Fig 1D). Instead, these findings collectively suggest the crucial role of MARylation removal in inhibiting SG assembly.

To investigate the impact of nsP3 expression on ADP-ribosylation levels of SG-associated mRNP complexes, we transfected cells with either GFP-vector or GFP-tagged nsP3, treated them with 0.2 mM arsenite, and performed polysome profiling fractionation to enrich SG-associated proteins (Ohn et al, 2008; Jayabalan et al, 2016) (Fig 1E). SG-associated translation factor RACK1 was enriched in monosome-containing fractions 4 and 5, indicative of translation arrest, as expected from arsenite treatment (Ohn et al, 2008). The samples were probed with a pan-ADP-ribose reagent to detect ADP-ribosylated proteins, irrespective of their forms. An accumulation of ADP-ribosylated proteins associated with fractions #2 to #6 was observed in control cells expressing the GFP vector (Fig 1E). In contrast, cells expressing GFP-tagged nsP3 exhibited a reduction in ADP-ribosylation signals in these fractions, including those enriched with SG-associated RACK1 (Fig 1E). These findings suggest that nsP3 expression inhibits SG assembly, likely by reducing ADP-ribosylation of target proteins.

### The MARylating enzyme PARP10 is critical for the initial stage of SG assembly

Given the strong inhibitory effect of MAR-degrading nsP3 on SG assembly, we hypothesized that MARylation is critical for SG assembly. To pinpoint the specific MAR-adding ADP-ribosyltransferase(s) involved in SG assembly, we systematically knocked down each enzyme within this subclass and assessed their impact on SG formation. As expected, following a 30-min arsenite treatment, cells transfected with the non-targeting siRNA control (siCon) exhibited SGs in ~90% of cells, as indicated by markers G3BP1 and eIF3b (Figs 2A and B and S1A). In this screen, nearly all of these knockdowns did not significantly affect the percentage of cells with SGs, except for PARP10 and PARP14. Knockdown of PARP10 and PARP14 reduced cells with SGs to 44% and 58%, respectively (Fig 2A and B). No synergistic effect was observed in SG inhibition under PARP10/PARP14 double knockdown, suggesting that these PARPs may work in the same pathway in SG assembly (Fig S1B and C). Consistently, a recent report showed that PARP14 regulates SG assembly in ovarian cancer cells (Challa et al, 2025). Given that PARP10 knockdown has a stronger effect, we then focused on elucidating the role of PARP10 in SG assembly. The effect of PARP10 knockdown on reducing the percentage of cells with SGs was confirmed with different siRNAs in U2OS cells and A549 cells (Fig S1D–G). Taken together, our findings

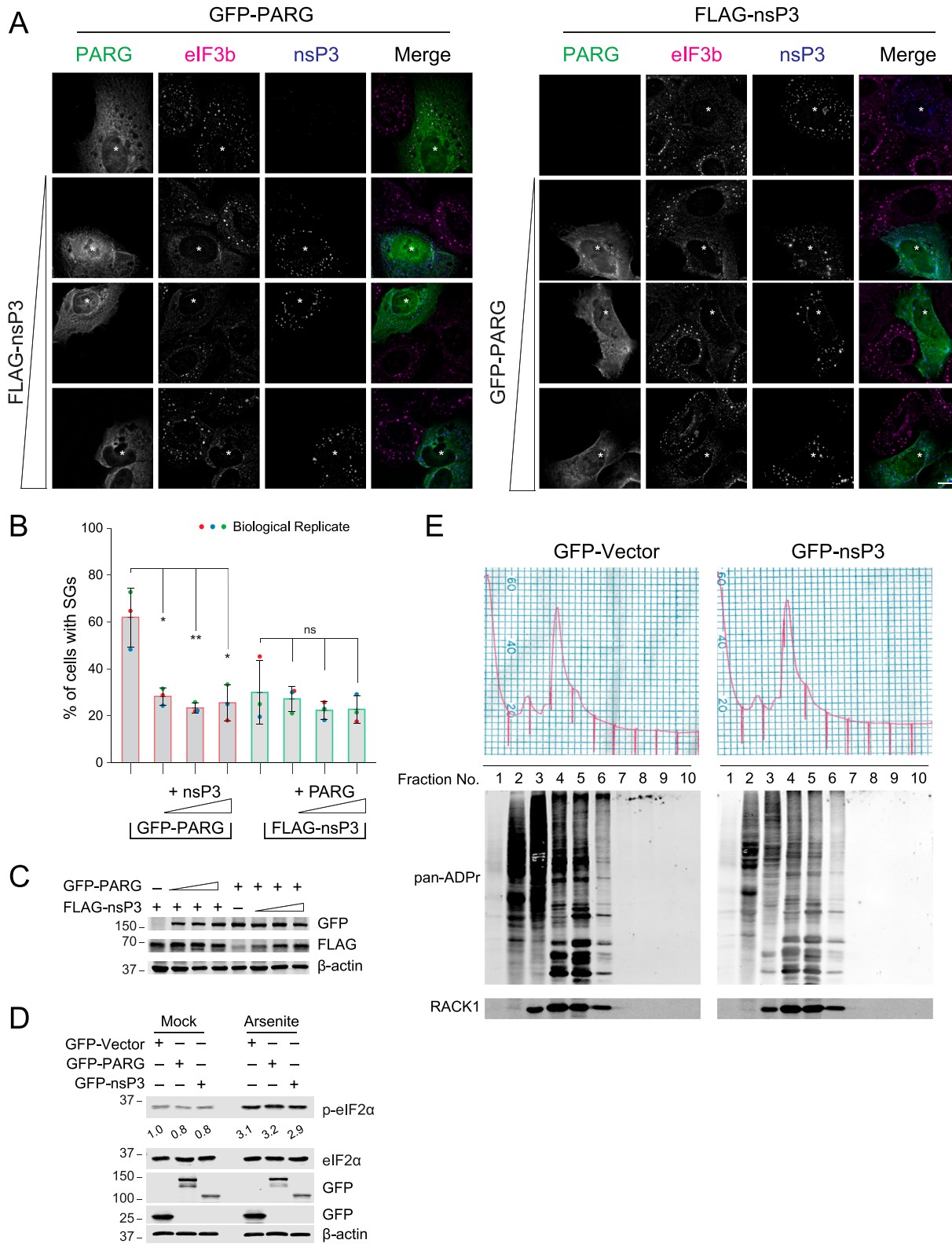

**Figure 1. Removal of terminal ADP-ribose is critical to inhibit SGs.**
**(A)** U2OS cells were transfected with FLAG-tagged nsP3 alone or co-transfected with increasing concentrations of GFP-tagged PARG (Right). In addition, U2OS cells were transfected with GFP-tagged PARG alone or co-transfected with increasing concentrations of FLAG-tagged nsP3 (Left). 24 h post transfection, cells were treated with 0.2 mM arsenite for 30 min, fixed, and immunostained for eIF3b and FLAG. Asterisk indicates co-transfected cells. Scale bar, 10 μm. **(A, B)** Bar graph showing the percentage of GFP/FLAG positive cells containing SGs in panel (A). Cells were scored as SG positive when eIF3b condensates are present in cells expressing both FLAG- and GFP-tagged constructs. Data represent mean ± SD from three biological replicates. **(A, C)** Western blot confirming the expression of transfected constructs in panel

suggest that SG assembly requires PARP10 and, to a lesser degree, PARP14.

Previously, we screened all human PARPs and identified the localization of PARPs 5a, 12, 13, 14, and 15 at SGs but not PARP10 (Leung et al, 2011). Given the pronounced effect of PARP10 knockdown on reducing the percentage of cells with SG, we re-examined its relationship with SGs. As reliable antibodies for staining endogenous PARP10 were lacking, we utilized GFP-tagged PARP10 to observe its cellular localization. Cells expressing PARP10 formed condensates distinct from SGs (Fig S2A), in agreement with previous studies showing that PARP10 is capable of forming condensates, regardless of the type or position of the protein tag (Kleine et al, 2012; Eckei et al, 2017; Mayo et al, 2018). Additionally, high PARP10 levels in cells inhibited SG formation (Fig S2A), consistent with previous findings that overexpressing PARP10 depletes NAD$^+$ (Heer et al, 2020), potentially hindering ADP-ribosylation that potentiates SG assembly.

To overcome this limitation, we employed a doxycycline-based system to minimally induce PARP10 expression (Fig 2C–E). This low-level expression did not affect SG assembly, with 87% of PARP10-expressing cells displaying SGs upon arsenite treatment—a rate comparable to that of untransfected cells. However, even at these expression levels, PARP10 remained in condensates distinct from SGs. We then explored whether this unique localization is tied to its catalytic activity. Using the PARP10 catalytic dead mutant G888W (Kleine et al, 2008), we observed that this mutant was still localized as condensates distinct from SGs (Fig 2C). Intriguingly, its expression, even at a lower level than that of the WT, significantly reduced the cells with SGs to only 36% (Fig 2E). Taken together, both excessive levels of WT PARP10 expression and the presence of the catalytically inactive mutant negatively impact SG assembly.

Endpoint-based assays, which only offer a static view of SG presence in cells, may not fully distinguish whether the reduced percentage of cells with SGs observed upon PARP10 knockdown is due to impaired SG assembly, accelerated disassembly, or enhanced degradation (Ohn et al, 2008; Kedersha et al, 2016). To investigate how PARP10 impacts SG kinetics, we generated a cell line that stably knocked down PARP10 and performed live cell imaging. Specifically, we employed lentiviral transduction to introduce PARP10 shRNA (shPARP10) into a cell line that endogenously tags the SG core component G3BP1 with GFP (GFP-ki-G3BP1; Fig 2F), allowing for real-time monitoring of SG assembly. Consistent with the transient knockdown results, the shPARP10 line showed a significant reduction of cells with SGs to 45% compared to the parental line (89% cells with SGs) after 30 min of arsenite treatment (Figs 2F and G and S2B). Live-cell imaging further revealed that SG assembly was notably delayed in the shPARP10 cell line relative to the parental line (Fig 2H and I, Video 1). As time progressed, SG assembly in both lines reached the same plateau by ~75 min (Fig 2H). These findings collectively underscore the crucial role of PARP10 in the initial stages of SG assembly.

## PARP10 is required for G3BP1-mediated SG assembly

G3BP1 and its paralog G3BP2 are jointly essential SG components, as G3BP1/2 double knockout (dKO) cells fail to form SGs upon various stress conditions, including arsenite treatment (Kedersha et al, 2016; Yang et al, 2020). However, reintroducing at least one of the G3BPs (either G3BP1 or G3BP2) restores SG formation in the dKO (Yang et al, 2020). A recent study further indicates that SG formation in G3BP1/2 dKO cells can also be restored by a synthetic construct mimicking G3BP1 (Yang et al, 2020). This mimic replaces G3BP1's essential domains responsible for SG assembly with analogous domains from other proteins (Fig 3A).

To investigate PARP10's role in G3BP1-mediated SG assembly, we first transfected the G3BP1/2 dKO cell line with either PARP10 siRNA or non-targeting siRNA (siCon). Subsequently, we transfected this cell line with GFP-G3BP1 to restore the core component necessary for SG assembly (Fig 3A and B). The reintroduction of GFP-G3BP1 restored SG formation in ~55% of cells transfected with non-targeting siRNA, as evidenced by the presence of eIF3b in GFP-G3BP1 condensates (Fig 3C and D). In contrast, only ~9% of cells with PARP10 knockdown exhibited SGs. These data suggest PARP10 is required for G3BP1-mediated SG assembly. Similarly, expressing the GFP-tagged G3BP1 mimic restored SG formation in ~66% of the G3BP1/2 dKO cells, confirmed by colocalization of eIF3b and GFP signals (Fig 3C and D). However, PARP10 knockdown reduced SG formation with the G3BP1 mimic to 29% (Fig 3C and D).

Our prior studies demonstrate a strong correlation between G3BP1 ADP-ribosylation and SG assembly: G3BP1 ADP-ribosylation increases under conditions when stress granules form, whereas this ADP-ribosylation reduces upon infection when the MAR-degrading nsP3 macrodomain inhibits SG assembly (Leung et al, 2011; Jayabalan et al, 2021). Indeed, when G3BP1 was reintroduced into the G3BP1/2 dKO cell line, leading to SG restoration, G3BP1 was found to be ADP-ribosylated (Fig 3E). This ADP-ribosylation was significantly reduced upon PARP10 knockdown (Fig 3F and G). Similarly, the GFP-tagged G3BP1 mimic was also ADP-ribosylated in a PARP10-dependent manner (Fig 3E–G). Taken together, these data suggest that PARP10 regulates SG assembly by facilitating ADP-ribosylation of G3BP1 or its functional mimics.

## SG core composition is altered upon PARP10 knockdown

While previous studies have underscored the significance of PAR in forming SGs by recruiting proteins through non-covalent interactions (Leung et al, 2011; Catara et al, 2017; McGurk et al, 2018; Duan et al, 2019), the involvement of PARP10—which adds MAR and not PAR (Kleine et al, 2008)—presents a conundrum. To probe deeper into this relationship, we analyzed the ADP-ribosylation form of G3BP1. By using antibodies that specifically detect either MAR or PAR (Bonfiglio et al, 2020; Weixler et al, 2023), we probed the GFP immunoprecipitates from G3BP1/2 dKO cells expressing GFP-

---

(A). **(D)** U2OS cells transfected with either GFP-vector, GFP-tagged nsP3 or PARG were either mock-treated or treated with 0.2 mM arsenite for 30 min and subjected to Western blot with indicated antibodies. Data represent results from two independent experiments. **(E)** U2OS cells either transfected with GFP-vector or GFP-tagged nsP3 were treated with 0.2 mM arsenite for 30 min and subjected to polysome profiling. Ten fractions collected from the sucrose gradient were prepared for Western blot analyses against pan-ADPr reagent. Data represent results from two independent experiments.
Source data are available for this figure.

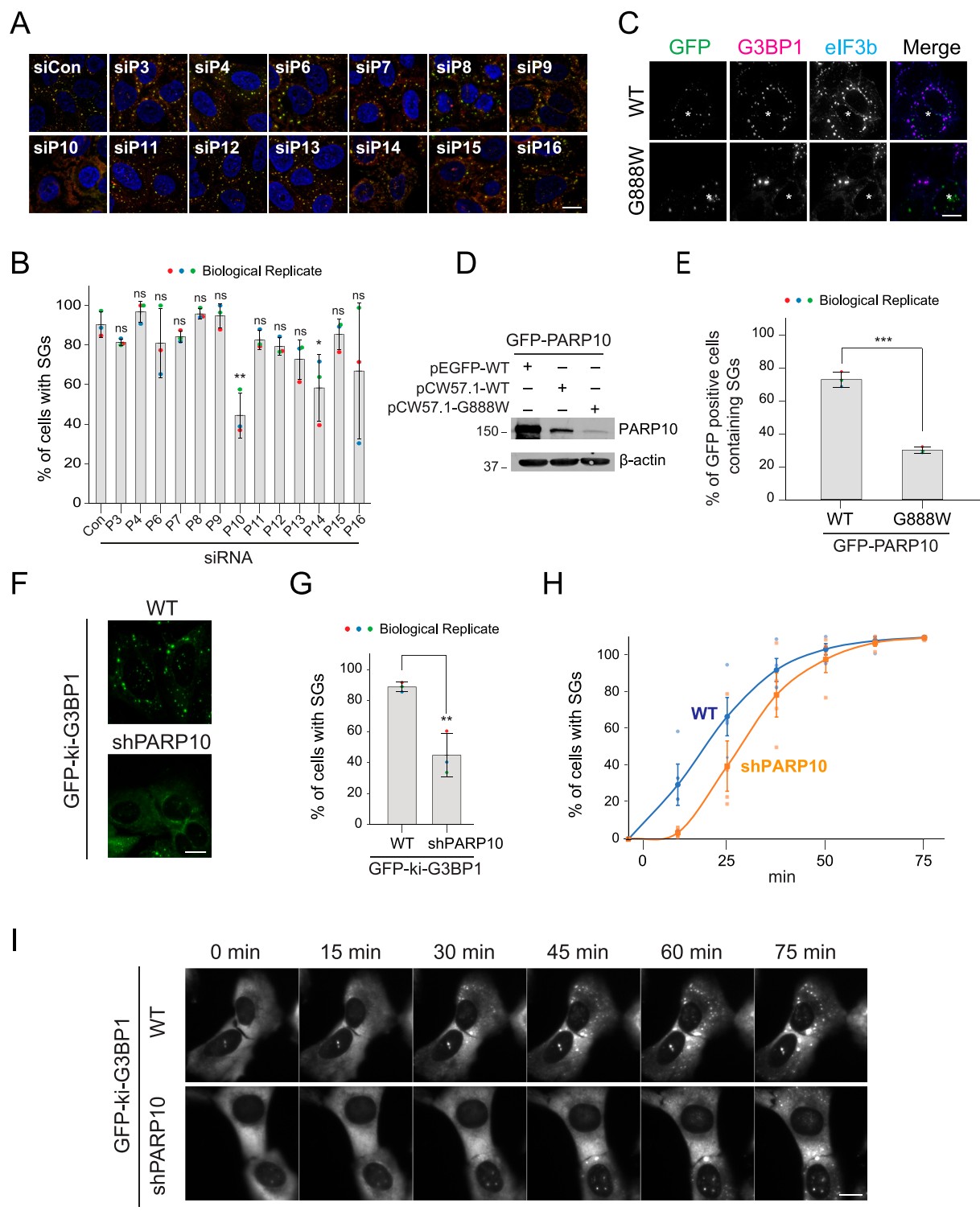

**Figure 2. PARP10 knockdown delays SG assembly.**
**(A)** U2OS cells were transfected with non-targeting siRNA (siCon) or siRNA against different PARPs as indicated. 36 h after transfection, cells were treated with 0.2 mM arsenite for 30 min, fixed, and immunostained for G3BP1 (green) and eIF3b (magenta). **(A, B)** Bar graph showing the percentage of cells with SGs in panel (A). Data represent mean ± SD from three biological replicates. **(C)** Dox-inducible GFP-tagged PARP10 WT or G888W mutant constructs were transfected into U2OS cells. After 12 h, cells were changed to fresh medium containing Dox and incubated for 24 h at 37°C. Cells were then treated with 0.2 mM arsenite for 30 min and fixed for immunofluorescence. **(D)** Expression level of constitutively active GFP-PARP10, dox-inducible GFP-tagged PARP10 WT and G888W mutant in U2OS cells. Dox was added for 24 h to induce the expression of PARP10 WT and G888W mutant. **(C, E)** Bar graph showing the percentage of cells containing SGs in panel (C). Data represent mean ± SD from three biological replicates. **(F)** GFP-ki-G3BP1 WT or shPARP10 stable cell lines were treated with 0.2 mM arsenite for 30 min and fixed for immunofluorescence. **(F, G)** Bar graph showing the percentage of cells with SGs in panel (F). Data represent mean ± SD from three biological replicates. **(H)** Graph showing the percentage of cells with SGs as a time series upon 0.2 mM arsenite treatment in WT and shPARP10 cell line. A total of 195 cells (GFP-ki-G3BP1 WT) and 336 cells (GFP-ki-G3BP1 shPARP10) from

G3BP1. To verify the specificity of these antibodies, we transfected GFP-tagged PARP1 and PARP10, which are present in PARylated and MARylated form, respectively (Lüscher et al, 2022). Intriguingly, our data indicated that G3BP1 is present in both MARyalted and PARylated forms (Fig 4A). We also observed that the G3BP1-mimic was also present in both forms (Fig 4B). Since ADP-ribosylation of G3BP1 is dependent on PARP10, it is plausible that PARP10 initiates the ADP-ribosylation of G3BP1, which can then be further PARylated by other PARPs present endogenously in cells.

Given that shPARP10 cells have reduced ability to assemble SGs (Fig 2H and I), we tested whether it reflects on the phosphorylation of eIF2$\alpha$—a critical determinant of SG assembly in cells treated with arsenite. Intriguingly, shPARP10 cells showed significantly lower induction of phosphorylated eIF2$\alpha$ levels at 15 and 30 min following arsenite exposure compared to shCon cells (Fig 4C and D). The reduced eIF2$\alpha$ phosphorylation level may partly explain the delay in initiating SG formation in PARP10 knockdown cells. To further test whether the SG defect in PARP10 knockdown cells is specific to eIF2$\alpha$-dependent assembly, we utilized Pateamine A, which induces SGs independently of eIF2$\alpha$ phosphorylation. While PARP10 knockdown reduced SG formation in arsenite-treated cells, it had no effect on SGs induced by Pateamine A, suggesting that PARP10 is likely involved upstream of eIF2$\alpha$-dependent SG formation (Fig S2C and D).

Considering the critical role of PAR in recruiting proteins to SGs (McGurk et al, 2018; Duan et al, 2019; Leung, 2020; Rhine et al, 2022), we hypothesized that reduced ADP-ribosylation under PARP10 knockdown condition might alter SG composition. To test this hypothesis, we employed a differential centrifugation method to biochemically isolate SG cores (Wheeler et al, 2017) from shPARP10 and its corresponding parental cell line after arsenite treatment. We then probed these cores with antibodies against several SG components (Jain et al, 2016). Given that several translation factors, such as eIF3g, bind to PAR (Jayabalan et al, 2021), we particularly focused on this class of SG components. Intriguingly, SG cores from the shPARP10 cell line exhibited an altered composition. Translation factors eIF3g, eIF3j, and eIF4AI/II were less enriched in SG cores compared to those from the parental line (Fig 4E). These data suggest that PARP10 modulates SG assembly at multiple levels, ranging from the regulation of eIF2$\alpha$ phosphorylation to the recruitment of translation factors within SGs.

# Discussion

Here, we pinpointed MARylation as a critical form of ADP-ribosylation integral to SG assembly. The MARylating enzyme PARP10 regulates the initial stage of SG assembly and targets the core component G3BP1. A decrease in PARP10 expression reduced eIF2$\alpha$ phosphorylation and the levels of translation factors within the SG core.

## MARylating enzymes drive PAR-enriched SG assembly

Many SG proteins are PARylated or bind PAR (Leung et al, 2011; Catara et al, 2017; Duan et al, 2019; Dasovich et al, 2021). Over-expressing PARG suppresses SG assembly, while its knockdown delays SG disassembly, indicating the critical role of PAR in maintaining SG integrity. However, inhibitors against PAR-adding PARPs, including the SG-localized PARP5a, do not suppress SG formation (Isabelle et al, 2012; Catara et al, 2017; McGurk et al, 2018; Duan et al, 2019). Instead, these inhibitors against PARP1/2 or PARP5a/b only block the localization of selective components to SGs, suggesting that PAR controls SG composition (Isabelle et al, 2012; Catara et al, 2017; McGurk et al, 2018; Duan et al, 2019). On the other hand, MARylation emerges as a key player in SG assembly. Three out of five PARPs localized in SGs are MAR-adding, and overexpression of any of them is sufficient to induce SG formation. Our systematic knockdown screen now further revealed that both PARP10 and, to a lesser extent, PARP14 are critical for SG formation. Unlike PARP14, PARP10 is not localized to SGs, suggesting that the MARylation responsible for SG formation can be initiated outside SGs.

In addition to the critical role of PARP10 in initiating SG assembly, a precise equilibrium in PARP10 levels is pivotal. Disruption of this balance—either by overexpression or by introducing a catalytically inactive mutant—compromises SG formation, even with endogenous PARP10 presence. These data suggest a dominant-negative effect, where the mutant interferes with the function of the WT PARP10 or other essential components of SG formation. One potential mechanism is that either an excess of PARP10 or its lack of catalytic activity might sequester essential substrates or cofactors required for SG formation. These findings underscore the nuanced balance required for PARP10's involvement in SG assembly, highlighting its critical regulatory function.

## The role of PARP10 and its substrates in SG assembly

G3BP1 is central to SG assembly, which is significantly ADP-ribosylated upon stress (Leung et al, 2011; Jayabalan et al, 2021). In this study, we identified G3BP1 as a substrate for PARP10. In G3BP1/2 dKO cells, reintroducing G3BP1 led to its ADP-ribosylation and restoration of SGs—both processes dependent on PARP10. Therefore, while G3BP1 is essential for SG assembly, its function may depend on PARP10. A surprising outcome of our work is that the synthetic G3BP1 mimic, which replaces the NTF2-like, RRM, and IDR modules with analogous domains from unrelated proteins (Yang et al, 2020) nevertheless requires PARP10. This observation suggests that PARP10 targets generic physicochemical features rather than a specific G3BP1 sequence motif. It also underscores the utility of minimalist scaffolds for dissecting the rules of condensate regulation: by substituting individual domains, we can test whether MARylation acts on RNA-binding modules, dimerization domains, or disordered tails for SG formation.

three biological replicates were analyzed. **(H, I)** Snapshot of live-cell imaging data from WT and shPARP10 cell line corresponding to panel (H). Asterisk indicates co-transfected cells. Scale bar, 10 $\mu$m.
Source data are available for this figure.

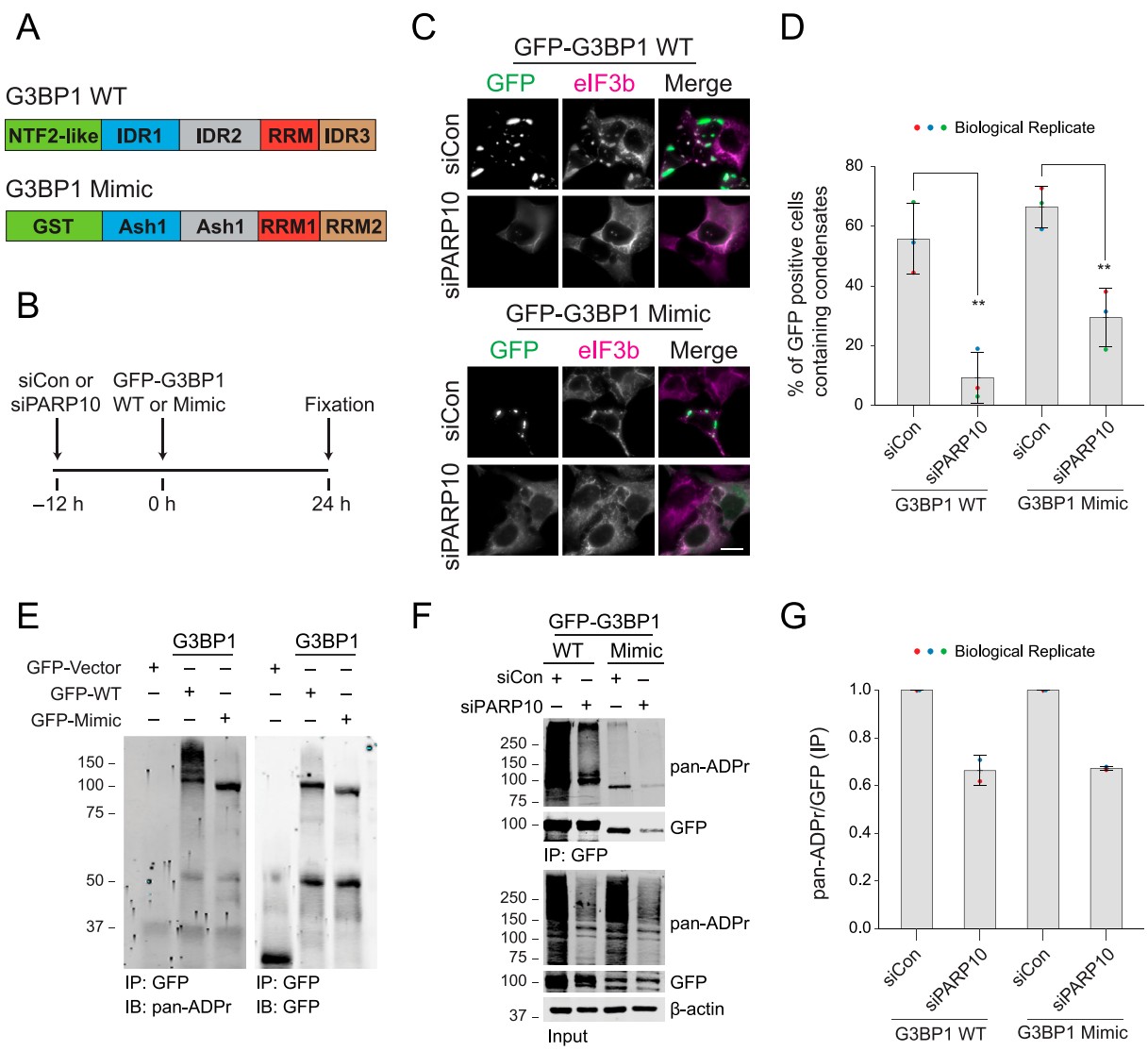

**Figure 3. PARP10-mediated ADP-ribosylation modulates SG condensation.**
**(A)** Domain structure of G3BP1 WT and mimic construct. **(B, C, F)** Experimental flow for panels (C, F). **(C)** G3BP1/2 dKO cells were either treated with siCon or siPARP10 for 12 h. After 12 h, cells were transfected with either GFP-tagged G3BP1 WT (Top) or G3BP1 mimic constructs (Bottom) for an additional 24 h. Post transfection, cells were fixed and immunostained for eIF3b. Asterisk indicates transfected cells. Scale bar, 10 μm. **(D)** Bar graph showing the percentage of GFP-positive cells containing SGs. Data represent mean ± SD from three biological replicates. **(E)** G3BP1/2 dKO cells were transfected with either GFP-vector, GFP-tagged G3BP1 WT or mimic constructs for 24 h. Post transfection, cells were immunoprecipitated using anti-GFP antibodies, and blotted against pan-ADPr reagent. **(F)** G3BP1/2 dKO cells were either treated with siCon or siPARP10 for 12 h, followed by transfection with either GFP-tagged G3BP1 WT or G3BP1 mimic constructs for 24 h. Post transfection, cells were lysed, immunoprecipitated, and blotted against pan-ADPr reagent. **(G)** Quantification of relative pan-ADPr band intensity between siCon and siPARP10 conditions. Data represent results from two independent experiments.
Source data are available for this figure.

Given that ADP-ribosylation is observed in multiple SG proteins, including TIA-1, TDP-43, and hnRNPA1 (Leung et al, 2011; McGurk et al, 2018; Duan et al, 2019), additional studies should examine their potential involvement in SG assembly. Alternatively, the mimic may emulate the native G3BP1's function, recruiting specific proteins for PARP10 modification. Future studies should determine if SG assembly is primarily driven by the covalent addition of ADP-ribosylation to core SG components or through non-covalent interactions by specific proteins that SG core components recruit through PAR. Regardless, this study emphasizes the crucial role of

ADP-ribosylation mediated by PARP10, a MAR-adding enzyme, to maintain the phosphorylation of the translation factor eIF2α and the presence of other translation factors within the SG core. These data suggest that PARP10 may mediate SG assembly at multiple steps, from modulation of stress signaling to stress granule condensation and composition. This observation aligns with our earlier findings, wherein a viral MAR-degrading enzyme was shown to disrupt the association between RNA-binding proteins and translation factors in SGs during infection (Jayabalan et al, 2021). Given the involvement of PARP10 in translation factor

**Figure 4. PARP10 initiates SG condensation and recruits SG components.**
**(A)** G3BP1/2 dKO cells were transfected with vectors encoding GFP, GFP-tagged G3BP1, G3BP2, PARP1, or PARP10 and after 24 h, immunoprecipitated using GFP-antibodies, and subjected to Western blotting against antibodies specific to MAR or PAR. Cells transfected with GFP-tagged PARP1 were subjected to 1 mM $H_2O_2$ treatment for 10 min to induce PARylation. **(B)** G3BP1/2 dKO cells were transfected with GFP-vector, GFP-tagged G3BP1 WT or mimic for 24 h. Cells were then lysed, immunoprecipitated using GFP antibodies, and subjected to Western blotting against antibodies specific to MAR or PAR. **(C)** GFP-ki-G3BP1 shCon or shPARP10 cells were subjected to 0.2 mM arsenite for 15 and 30 min, and cells were lysed to blot against indicated antibodies. **(D)** Quantification of p-eIF2α/total eIF2α intensity in

localization, its role in translation regulation warrants further investigation.

### Working model: MARylation as a rate-limiting step of SG assembly

Building on these observations, we propose that MARylation of one or more substrates is crucial for SG assembly. Data from PARP10 or PARP14 knock down points towards potential redundancy in the MAR-adding mechanism. These MARylated substrates likely act as anchors, facilitating protein interactions that initiate SG assembly. Moreover, PAR-adding PARP5a or PARP1 might extend this MARylation to PARylated forms for additional protein interactions through this polynucleotide-like polymer (Fig 4F). In line with this model, we identified a PARP10-dependent modification of G3BP1 as both MARylation and PARylation.

One scenario is that PARP10 mediates the initial stage of SG assembly through ADP-ribosylation of G3BP1 as well as by regulating eIF2$\alpha$ phosphorylation. This action, combined with the inherent RNA and protein-protein interactions of G3BP1 (Kedersha et al, 2016; Guillén-Boixet et al, 2020), drives the condensation process. Consistent with prior studies, G3BP1 predominantly interacts with the same protein partners even in the unstressed state (Markmiller et al, 2018). However, upon stress conditions, the concentration of G3BP1 and these interacting proteins is likely to increase and facilitate condensation (Jain et al, 2016; Markmiller et al, 2018). We postulated that ADP-ribosylation, in addition to RNA, enhances G3BP1's interactions with proteins via PAR. This positions ADP-ribosylated G3BP1 as a super-scaffold within SGs (McGurk et al, 2018; Duan et al, 2019; Dasovich et al, 2021; Jayabalan et al, 2021), bridging proteins through interactions with two nucleic acid polymers.

Notably, both PARP10 and PARP14 possess RNA-binding domains, hinting at a deeper interplay with RNA dynamics in PAR-enriched SG assembly. Our recent work demonstrates that even a 1: 1,000 sub-stoichiometric ratio of PAR can potently condense the SG-localized RNA-binding protein FUS (Rhine et al, 2022). While FUS co-binds and co-condenses with RNA and PAR, RNA can also enter pre-formed FUS-PAR condensates, displacing a majority of the existing PAR (Rhine et al, 2022). These findings, along with the kinetic role established here for ADP-ribosylation in SG assembly, led us to propose a model: SG assembly is initiated by the rate-limiting step of MARylation—both by modifying substrate proteins and by maintaining p-eIF2$\alpha$ levels—followed by the formation of PAR, which acts as a catalyst for protein condensation. This initial phase of transient PAR-protein interactions then gives way to more stable protein-RNA interactions, culminating in SG formation.

### Limitations

Our data point to PARP10 as the primary PARP involved in SG assembly. However, it is possible that different cell lines might depend on other MAR-adding enzymes, such as PARP14 (Zhen et al, 2017; Challa et al, 2025). While the effects of the catalytic dead mutant highlight the significance of PARP10 activity, we cannot conclusively determine if the observed knockdown effects are due to its physical absence or its enzymatic function. While our study is centered on SG assembly, the relative significance of MARylation versus PARylation remains unclear for SG disassembly. Our previous data indicate that knocking down PARG, resulting in increased PAR levels, prolongs SG disassembly. Conversely, the catalytic mutant of the viral MAR-degrading enzyme delays SG disassembly during infection. Exploring SG disassembly may reveal novel therapeutic strategies targeting SGs or SG-like condensates in viral infection, cancer, and neurodegeneration.

## Materials and Methods

### Cell culture, chemicals, and transfection

U2OS and A549 cells were obtained from the American Type Culture Collection (ATCC). Cells were maintained in DMEM (Gibco) containing 10% heat-inactivated FBS (Gibco, Life Technologies). Plasmids were transfected using JetPrime from Polyplus as per the manufacturer's protocol. Stress granules were induced using 0.2 mM sodium arsenite or 100 nM Pateamine for 30 min.

### Immunoblot analysis

Cells were lysed in a RIPA buffer (50 mM Tris–Cl pH 8.0, 150 mM NaCl, 0.1% sodium dodecyl sulfate [SDS], 1% Nonidet P-40, 1 mM EDTA with phosphatase and protease inhibitor cocktails) with 10 $\mu$M Olaparib (PARP inhibitor) and 10 $\mu$M PDD 00017273 (PARG inhibitor) on ice for 15 min, followed by centrifugation at 18,000$g$ for 15 min at 4°C. Protein samples were acetone-precipitated for at least 1 h at –20°C. Precipitates were centrifuged at 16,000$g$, 4°C for 15 min, and the air-dried pellets were then diluted in 1 × SDS sample buffer. The samples were resolved in polyacrylamide gel electrophoresis and blotted with appropriate primary antibodies. All primary and secondary antibodies used in this study are listed in Table S1.

### Doxycyline induction

U2OS cells (~3 × $10^5$) were first transfected with respective constructs and allowed to grow for 12 h. After 12 h, medium was replaced with fresh DMEM containing 2 $\mu$g/ml (for WT) and 4 $\mu$g/ml (for G888W) doxycycline and incubated for additional an 24 h before arsenite treatment.

---

panel (C). Data represent mean ± SD from three biological replicates. **(E)** GFP-ki-G3BP1 shCon and shPARP10 cells were treated with 0.2 mM arsenite for 30 min. Following stress induction, SGs were isolated, and subjected to Western blot analyses against indicated antibodies. **(F)** Proposed model of PARP10-mediated G3BP1 MARylation and further extension of ADP-ribose chain by other PARPs.
Source data are available for this figure.

## Generation of shPARP10 stable cell line

U2OS cell line endogenously expressing GFP-tagged G3BP1 (GFP-ki-G3BP1, a kind gift from Dr. Paul Taylor) was used to generate shPARP10 stable cell line. Briefly, lentiviruses were prepared using either PARP10 shRNA clone (TRCN0000052947; Sigma-Aldrich) or non-targeting shRNA control (SHC202V), and used to transduce the GFP-ki-G3BP1 cells. Transduced cells were selected with puromycin, and the pooled population of puromycion-resistant cells were screened by RT-qPCR to confirm PARP10 knockdown efficiency.

## Immunofluorescence

U2OS cells (~6 × 10^4) grown on coverslips were treated with 0.2 mM arsenite for 30 min. Post-treatment, cells were washed twice with 1 × PBS, fixed with 4% PFA for 15 min at RT, permeabilized with ice-cold methanol for 10 min, and washed twice with 1 × PBS. Cells were then blocked with 5% normal horse serum in 1 × PBS containing 0.02% sodium azide for 1 h at RT. All primary antibodies were diluted in blocking buffer and incubated either 1 h at RT or overnight at 4°C, followed by three washes with 1 × PBS, 10 min each. Finally, cells were incubated with appropriate secondary antibodies diluted in blocking buffer (1:500) for 1 h at RT, washed three times with 1 × PBS, and the coverslips were mounted on glass slides using Prolong Gold. All primary and secondary antibodies used in this study are listed in Table S1.

## Immunoprecipitation

U2OS WT or G3BP1/2 dKO cells were lysed in cold lysis buffer (CLB) (50 mM Hepes pH 7.4, 150 mM NaCl, 1 mM MgCl$_2$, 1 mM ethylene glycol-bis($\beta$-aminoethyl ether)-N,N,N',N'-tetraacetic acid, 1% Triton X-100 supplemented with 1 mM NaF, 1 mM PMSF, 10 $\mu$M Olaparib, and 10 $\mu$M PDD 00017273). Cell lysates were mixed at 4°C for 15 min, spun down for 15 min at 16,000$g$, and the supernatant fluid was collected in a new tube. The GFP-magnetic bead complex was prepared by incubating anti-GFP (3E6; Invitrogen) with DYNA magnetic beads (10004D; Invitrogen) at RT for 10 min. The complex was then washed once with lysis buffer and the cleared lysates were added. Samples were incubated for 2 h at 4°C. Beads were washed once with CLB, twice with high-salt CLB (300 mM NaCl), followed by a final wash with CLB. The precipitates were then boiled with 1 × SDS sample buffer for 10 min at~85°C. All experiments were performed at least thrice.

## Polysome profiling analysis

Polysome profiling analysis was carried out as described previously (Jayabalan et al, 2016). Briefly, U2OS cells were transfected with GFP-vector, or GFP-tagged nsP3 were treated with 0.2 mM arsenite for 30 min. Following 30 min, 10 $\mu$g/ml cycloheximide was added to the cells and incubated for 5 min at RT, washed with cold PBS, lysed with 500 $\mu$l polysome lysis buffer (20 mM Hepes pH 7.6, 5 mM MgCl$_2$, 125 mM KCl, 1% NP-40, 2 mM DTT) supplemented with 100 $\mu$g/ml cycloheximide, SUPERase-In, 1 mM NaF, 1 mM PMSF, 10 $\mu$M Olaparib and 10 $\mu$M PDD 00017273 at cold room. Cell lysates were tumbled for 15 min at 4°C and centrifuged at 16,000$g$ for 15 min. The supernatants were fractionated in 17.5–50% linear

sucrose gradients by ultracentrifugation (217,300$g$ for 2 h 40 min) in a Beckman ultracentrifuge using SW40-Ti rotor. Gradients were eluted with a gradient fractionator (Brandel) and monitored with a UA-5 detector (ISCO). Fractions were acetone precipitated at −20°C for overnight and the precipitates were then boiled with 1 × SDS sample buffer for 10 min at~90°C.

## Live-cell imaging

GFP-ki-G3BP1 WT and shPARP10 cells were seeded in two-well–chambered cover glass (Nunc Lab-Tek) at around 70% confluency. 1 h before imaging, cells were pre-incubated with CO$_2$-independent medium containing 20% FBS. Cells were incubated with 0.2 mM arsenite and live-cell imaging was performed simultaneously for WT and shPARP10 cell lines. A total of 15–20 random fields were chosen and imaged, and the presence of SG was monitored as microscopically visible condensates in the FITC channel. Images were acquired at 5-min intervals for 90 min, controlled by the SoftWorx suite (GE Healthcare). Cells were imaged using a DeltaVision Elite system (GE Healthcare) microscope equipped with ×40 (1.516 N.A. oil) immersion objectives, a high-speed CCD Camera (Cool SNAP HQ2), appropriate filter sets for FITC, and an incubation chamber (37°C and 80% humidity). Presence of SG at each time point for WT (n = 195) and shPARP10 (n = 336) in three biological triplicates was analyzed using ImageJ software.

## RNA isolation and RT-qPCR analysis

Total RNA isolation was extracted using TRIZOL (Thermo Fisher Scientific) as per manufacturer's protocol. Briefly, cells were lysed and phase-separated with chloroform, precipitated with isopropyl alcohol, and washed with 70% ethanol twice. The pellet was then resuspended in ultrapure distilled water, and the quality/quantity was measured by NanoDrop (Thermo Fisher Scientific). Reverse transcription was performed with 1 $\mu$g RNA and random hexamers using the SuperScript VILO cDNA synthesis kit (Thermo Fisher Scientific). RT-qPCR was then performed using SYBR Green master mix using the primers tabulated, and the results were normalized by RPLP0. Each sample was run in technical triplicate.

## Isolation of stress granule cores

The core isolation was performed according to a previously described method (Jain et al, 2016; Wheeler et al, 2017). Briefly, GFP-ki-G3BP1 or GFP-ki-shPARP10 cell lines in 15 cm tissue-culture dishes (six plates per condition) were treated with 0.2 mM sodium arsenite for 30 min. After treatment, cells were washed once with cold PBS, scraped in PBS, and flash frozen with liquid N$_2$ and the cell pellets were stored at −80°C. The pellets were thawed on ice for 5 min and re-suspended in 200 $\mu$l stress granule lysis buffer (50 mM Tris–HCl, pH 7.4, 100 mM potassium acetate, 2 mM magnesium acetate, 0.5 mM DTT, 50 $\mu$g/ml heparin, 0.5% NP40, 1:5,000 Antifoam B, 1x EDTA-free protease inhibitor cocktail and 1:500 RNase inhibitor). To efficiently lyse the cells, 10 passages through the 25G gauge 5/8 syringe were performed. After lysis, spin the samples at 1,000$g$ for 5 min at 4°C to pellet cell debris, and the supernatants were transferred to new tubes and centrifuged at 18,000$g$ for 20 min at

4°C. The supernatants were discarded, and the pellets were resuspended with 1 ml stress granule lysis buffer. The samples were again centrifuged at 18,000$g$ for 20 min at 4°C. The pellets were re-suspended in 60 $\mu$l stress granule lysis buffer followed by spin at 850$g$ for 2 min at 4°C, and the supernatants representing the stress granule enriched fractions were boiled with 1x SDS sample buffer for 10 min at ~90°C.

### Stress granule quantification

We classified cells as SG-negative if the entire cell lacked both G3BP1 and eIF3b condensates. For quantitation, random 40× fields were chosen and a total of between 80 and 120 cells were counted per condition. For quantification of SGs from the live-cell imaging experiments, the number of SGs present at each time was calculated and plotted. In experiments involving the G3BP1/2 dKO cell line, cells were scored as SG positive when the GFP-expressing cells contained eIF3b condensate. All experiments were repeated three independent times.

### Statistical analysis

Data were presented as mean ± SD and groups were compared using the two-tailed unpaired $t$ test and the Kolmogorov–Smirnov test. All experiments represent at least three biological replicates unless mentioned. $P < 0.05$ was considered statistically significant.

## Data Availability

All data generated and analyzed in this study are included in this publication as Source Data files. Any remaining information can be obtained from the corresponding author upon reasonable request.

## Supplementary Information

## Acknowledgements

We thank members of the Leung laboratory for their critiques of the manuscript. We thank Dr. Nancy Kedersha and Dr. Paul Anderson for G3BP1/2 dKO cells and G3BP1 constructs, and Dr. Paul Taylor for GFP-ki-G3BP1 cell line. The authors would also like to thank Dr. Lee Kraus and colleagues for sharing their related manuscript prior to their submission. This work was supported by a Johns Hopkins Sol Goldman pancreatic cancer research center pilot project grant (AKL Leung) and NIH grant R01GM104135 (AKL Leung)

## Author Contributions

AK Jayabalan: conceptualization, data curation, formal analysis, investigation, methodology, and writing—original draft, review, and editing.

K Bhambhani: data curation, formal analysis, investigation, and methodology.

AKL Leung: conceptualization, formal analysis, supervision, funding acquisition, investigation, project administration, and writing—original draft, review, and editing.

### Conflict of Interest Statement

The authors declare that they have no conflict of interest.

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
