## [Reviewer comments · Life Science Alliance]

PARP10 is Critical for Stress Granule Initiation

Aravinth Jayabalan, Krishna Bhambhani, and Anthony Leung

DOI: <https://doi.org/10.26508/lsa.202403026>

Corresponding author(s): Anthony Leung, Johns Hopkins University

Review Timeline:

Submission Date:	2024-09-01
Editorial Decision:	2024-09-27
Revision Received:	2025-08-03
Editorial Decision:	2025-08-25
Revision Received:	2025-09-16
Editorial Decision:	2025-09-19
Revision Received:	2025-09-20
Accepted:	2025-09-24

Scientific Editor: Tim Fessenden

Transaction Report:

September 27, 2024

Re: Life Science Alliance manuscript #LSA-2024-03026-T

Dr Anthony K.L. Leung
Johns Hopkins University
Biochemistry and Molecular Biology
615 N. Wolfe Street, W8041
Baltimore, MD 21205

Dear Dr. Leung,

Thank you for submitting your manuscript entitled "PARP10 is Critical for Stress Granule Initiation" to Life Science Alliance. The manuscript was assessed by expert reviewers, whose comments are appended to this letter. We invite you to submit a revised manuscript addressing the Reviewer comments.

Thank you for this interesting contribution to Life Science Alliance. We are looking forward to receiving your revised manuscript.

Sincerely,

B. MANUSCRIPT ORGANIZATION AND FORMATTING:

Reviewer #1 (Comments to the Authors (Required)):

The manuscript by Jayabalan et al. reports on the crucial role of the ADP-ribose polymerase PARP10 for regulation of stress granules (SGs). The authors first investigated whether overexpression of the mono-ADP-ribose (MAR) hydrolase nsP3 or the poly-ADP-ribose (PAR) hydrolase PARG have an impact on SG number when overexpressed in U2OS cells. They found a greater reduction in SGs upon expression of the MAR hydrolase compared to the PAR hydrolase, implicating MAR removal as an important step in SG dynamics. Next, they screened 13 MAR-adding enzymes from the PARP family (using siRNA-mediated knockdown) for a potential effect on SG number, which identified PARP10 as major and PARP14 as additional hit. They further followed up the PARP10 finding. Using live cell imaging, they confirmed that shRNA-mediated knockdown of PARP10 causes a delay in the formation of G3BP1-positive granules, however GFP-tagged PARP10 does not localize to SGs, but forms distinct cytoplasmic granules. In G3BP1/2 double-KO cells, SG formation can be rescued by overexpressing GFP-G3BP1 (or a synthetic mimic), which is dependent on PARP10. The authors then seek to demonstrate PARP10-dependent MAR/PARYlation of G3BP1 (or the synthetic G3BP1 mimic), and that PARP10 knockdown alters SG composition and reduces phospho-eIF2alpha phosphorylation. They propose that MARYlation of G3BP1 (or other SG proteins) by PARP10 is a key first step in SG initiation, followed by PARYlation by other PARPs, e.g. PARP5a or PARP1. The polynucleotide-like polymer then may recruit further proteins and help in the assembly of SGs.

The study nicely complements prior work implicating MAR/PAR are crucial regulators of SG dynamics and highlights a so far unknown role of PARP10 in this process. It is therefore of interest to a wide cell biology/biochemistry readership. However, the study has several technical shortcomings and could benefit from some text/figure revisions to make it more understandable and technically sound.

Major points:

1. Tone down overstatement: On page 4, when describing their results, the authors several times mention that "MARYlation or nsP3 expression inhibits SG assembly" or that PARP10/14 knockdown results in "reduced SG assembly". This should be rephrased, as they simply demonstrate reduced SG numbers upon these treatments, however at this point it cannot be claimed that the reduced SG number is due to impaired SG assembly - it might as well be caused by accelerated SG disassembly or enhanced SG degradation. Subsequent experiments (Fig. 2/3) hint at this being indeed correct, but to not oversell their data, the reduced steady state number of SGs should be interpreted with more caution/ neutrality.
2. Figure 2I: The GFP-G3BP1 signal in the bottom row (shPARP10 line) looks much dimmer than in the upper row (WT line) - if this is a true phenotype, this is a severe concern, because SG formation is influenced by the G3BP1 concentration. So a cell line with lower levels of GFP-G3BP1 would show fewer SGs simply due to the lower protein levels and then the effect might not come from the PARP10 knockdown. To clarify this concern, the authors should demonstrate that both cell lines have equal levels of GFP-G3BP1 and if so, show more suitable snapshot images where both cell lines have (roughly) equal levels of GFP-G3BP1 and hence can be truly compared.
3. Methods details on how the GFP-ki-G3BP1 line and stable / dox-inducible GFP-PARP10-lines were generated are missing completely. This should be included in the methods sections, and experimental details (e.g. selection media, dox concentration) should be given. Similarly, methods details (qPCR) how the PARP knockdown efficiencies (shown in Fig. S1A and S2B) were determined are missing.
4. Fig. 2C: As GFP-tagging often alters a protein's condensation/localization behavior, the authors should examine PARP10 fused to a small epitope tag (ideally fused to the N- or C-terminus to exclude tagging artifacts), to validate the (punctate, but not SG-associated) localization seen for the GFP-tagged construct.
5. Figure 2D/E: A non-PARP10 overexpressing parental control is missing in both panels (particularly relevant would be to show in Fig. 2E what % of untransfected cells had SGs). Another major concern is that the catalytically inactive mutant (G888W) is expressed at much lower levels than the WT, hence the observed effect (fewer SGs in mutant-expressing line) could also be due to the lower expression levels. Hence, the authors should repeat the experiment ensuring equal expression or WT and G888W mutant (if necessary by adjusting the DOX concentration to achieve equal expression). If it is not possible to achieve this, the experiments cannot be properly interpreted and should be removed from the manuscript.
6. Figure 3E/F: GFP-IP should be performed under denaturing conditions (e.g. using 1% SDS in the lysis buffer + heating, followed by dilution into a native buffer that allows GFP-IP), otherwise one cannot be sure that the identified signals really come from direct modification of G3BP1 or the mimic (it could be from co-precipitated modified proteins that run at the same MW).
7. Figure 3E/F: The G3BP1-mimic has a strikingly different band pattern in the PAR blot (single band) compared to the high MW smear seen for G3BP1 WT, yet the authors do not comment on where this difference (likely) comes from. Is the G3BP1-mimic

MARylated instead of PARylated (have they reprobated the mimic-IP with a MAR-specific antibody?) The authors should make an effort to clarify what the difference in ADP-ribosylation pattern between the WT and mimic is, or at least point out the striking difference and speculate what the molecular difference could be. If the G3BP1-mimic is indeed MARylated instead of PARylated, this would be an interesting finding, hence this possibility should be explored and either validated or disproved.

Minor points:

- Page 3, line 37: mRNAs are never truly "naked" in the cell (still RBP-bound), rephrase.
- Page 3, line 40/41: Last sentence ("Yet,...") is very generic and in my view just disrupts the flow, so I recommend to delete it.
- Page 3, line 59: in addition to ref 24, ref 31 and 39 would be relevant and fitting citations here.
- Page 3, line 48: When talking about PARPs and PARG as known SG components, the authors could check whether these factors are also found in published SG proteomes / proximity maps (e.g. Jain et al., Cell 2015; Youn et al., Mol Cell 2018) and potentially mention this here (PMID: 26777405; PMID: 29395067).
- Page 4, lines 79: The authors state that PARG and nsP3 are co-expressed "at different expression levels". This statement is misleading, as the IF images in Fig. 1A show roughly equal expression levels across the titration. Presumably, as the DNA is titrated up, simply the % of transfected cells increases (hence the more prominent band in the Western blot for higher DNA conc.), however on the single cell level the expression level is likely constant. In line with this, there is not really a dose-dependent effect observed (quantification in Fig. 1B). To not mislead the reader, the statement about the "different expression levels" should be rephrased, or the titration could be omitted altogether and just 1 DNA concentration could be shown...
- Page 4, lines 95-97: Check grammar, sounds like a verb is missing here...
- Figure 1 would be more readable if the text and figure order were aligned (titration of flag-nsP3 described first in the text, but shown second in Fig. 1A and B - it would be more readable if they showed the nsP3 titration/quantification panel first, and the GFP-PARG titration data second).
- Similarly, Fig. 3 would be more readable/understandable if the G3BP1 mimic (shown throughout all panels) would be mentioned from the beginning of this results paragraph, along with G3BP1 WT, as it is quite confusing to always see this mimic in all figure panels, but not know why it's there or what it is. This confusion could be avoided by rewriting the results text and by introducing the mimic together with G3BP1 WT (on p. 7 already). Moreover, it should be pointed out why this mimic has been used and what can be learnt from it (not entirely clear to me because too little info on the mimic is given).
- Figure 1E, are the fraction numbers mislabelled? In figure 1D, fraction no.1 has no ADPr signal, however in 1E there is a strong ADPr signal in fraction no. 1. Correct or explain the discrepancy.
- Figure 1E: Since no quantification or statistics are shown: Authors should state in the legend how often this experiment was performed and that the shown WB is representative of n= ? experiments.
- Fig. 2D labelling is hard to understand, it took me a long time to find out what they mean with "WT" better label as "CONST. WT" (as opposed to DOX-inducible WT or mutant....)
- For live cell imaging (methods): mention the number of cells that were imaged/analyzed.

Reviewer #2 (Comments to the Authors (Required)):

Re: Jayabalan et al., "PARP10 is critical for stress granule initiation"

This is an investigation of the influence of ADP-ribosylation on formation of stress granules (SGs). The enzymes adding ADP-ribosyl units to target proteins are called poly-ADP-ribosyltransferases (PARPs). Despite their name only some of these add multiple ADP-ribosyl monomers to the target protein; others add only one subunit (mono-ADP-ribosylation; MARylation).

Here, the authors found that knockdown of PARP10, and to a lesser degree PARP14, results in a reduction and delay of SG formation. No other PARP knockdown caused a noticeable defect. The well-studied SG component G3BP1 was established as one of the substrates for PARP10. Based on previous comparisons between the effects on SG formation by a viral poly-ADP-ribosyl degrading enzyme and its cellular counterpart and experiments in the current manuscript, the authors inferred that addition of the first ADP-ribosyl unit (MARylation) is critical for SG assembly. The authors further demonstrate that G3BP1 is mostly PARylated, leading to a model where this protein is first MARylated by PARP10, and then poly-ADP-ribosylated, building on that first ADP-ribosyl unit by other PARPs.

Overall, the manuscript is of high technical quality and the claims are supported by experimental data. It is of interest to a wide scientific community. There are however some questions that should be addressed, see below.

- One obvious question that could have been answered within the scope of this manuscript is what happens with a double knockdown of PARP10 and PARP14. Given the authors' working model, where addition of the first ADP ribosyl subunit (MARylation) is a starting point for SG assembly, this would be suspected to result in a strong effect. This would be an interesting addition to the paper, and should not be too demanding to perform. The implications of a PARP10/PARP14 double knockdown could be particularly interesting for the model of SG formation since PARP14 is an SG component but PARP10 is not (line 256). Is it possible that ADP-ribosylation both within and outside of SGs are required for SG assembly?
- The authors used arsenite to induce SGs. The protein composition of SGs is affected by the specific stress used to induce

them. Is it possible that the requirement of ADP ribosylation for SG assembly is also affected by the type of stress?

- Another question is this: if PARP10 has such a fundamental role in SG assembly, how come it has escaped detection in previous screens, genome-wide or directed, for factors required for SG formation? There have been a number of these over the years, e.g. PMID 18794846, PMID 32393832. Was PARP10 for some reason not included in previous screens? Or is the effect on SG formation too limited to be reliably detected in a high-throughput format?

Minor remarks:

- The authors constructed a synthetic protein designed to mimick G3BP1 on an overall structural level despite having no primary sequence similarity, and showed that it also functions as a target of PARP10. However, we are not offered much in terms of a rationale why this experiment was performed in the first place, nor how its outcome should be interpreted. The authors claim that "This finding raises the intriguing possibility that ADP-ribosylation of any core SG component could be sufficient to initiate SG assembly, irrespective of its identity", but there is no evidence to support that idea.

- Line 166: "G3BP1 and its paralog G3BP2 are essential SG components" should be "...are synthetically essential..."

Reviewer #3 (Comments to the Authors (Required)):

The submitted work focuses on the role of PARP10 in SG formation. The lab is pioneer in the subject and had previously reported compelling evidence that parylation influences SG condensation. Here it's proposed that marylation (the addition of a single ADP ribose) by PARP10 is a rate-limiting step in the initiation of SG formation.

Experiments are in general technically well performed. However I don't fully agree with a number of rationales and interpretation, as follows.

1. A major concern relates to the potential pleiotropic effects of PARP10 loss-of-function. When reading the MS, from the very beginning, I felt that an important issue was missing, and is whether stress sensing was affected by PARP10 KD. Marylation is a constitutive post-translational modification and a yet unknown number of proteins can potentially be affected. In addition, as authors explained, ADP-ribosylation is connected to NAD metabolism, and the effect of PARP10 KD in the overall cellular fitness is unknown.

Acute stress sensing is greatly influenced by cell health. Preconditioning by previous stress insults, either chronic or acute, may reduce the stress response, with lower eIF2alpha phosphorylation and lower SG formation in several examples.

Thus, a main question is whether the strong reduction in the phosphorylation of eIF2a (down to 50 % in figure 4D) is the main cause of defective SG formation upon PARP10 KD.

A) A potentially useful experiment to solve this important issue is to attempt to induce SGs that form independently of eIF2a phosphorylation. Mitochondrial inhibitors may work. Also, translation initiation inhibitors such as hippuristanol were shown to induce SGs independent of eIF2a phosphorylation.

B) In the same line, is the arsenite-induced translational silencing affected by PARP10 KD? (this can be easily assessed by puromycilation)

C) Levels of arsenite-induced phospho 2alpha should be controlled upon PARG and nsP3 transfection, to allow the conclusion that the effect is downstream of stress sensing. (I don't recommend to assess translation by puromycilation in these cases, as overexpression of transfected proteins seriously interfere with the assay)

2. Figure 3E, F. Both G3BP-GFP and G3BPmimic-GFP are marylated by PAR10. Was this expected? Are the amino acids involved known in each case? Should we expect that any overexpressed SG protein will be parylated? Would it be possible to include a control of a transfected protein that won't be parylated?

3. Figure 4B. PARP10 KD reduces SG condensation, so that less G3BP-GFP is expected to be recovered in the pellet. I understand that the loading was adjusted to have equal amounts of G3BP-GFP (first row). Simultaneously, several initiation factors normally present in SGs are less abundant relative to G3BP. How is this interpreted? G3BP condensates contain less abortive translation initiation complexes? Is this indicative of defective translational silencing (see above)?

In summary, the model proposed in Fig4E (as well as the title and abstract) largely overstates the relevance of G3BP marylation in driving SG formation. Instead, the upstream defective eIF2a phosphorylation may fully explain the effect. If the target residue(s) in G3BP can be identified, a point mutation that abrogates marylation might directly assess the role of G3BP marylation.

MINOR:

1). Figure 1D (the effect of MAR-degrading nsP3) and line 98. " a reduction (of PAR signal) in fractions enriched with SG-

associated proteins". It appears that all lanes with significant OD (from 2 to 6) have reduced ADPribose signal. (Should we expect an effect specific of SG components?)

2) Figure 1E. The apparent change in pattern of lines 3 and 4 upon nsP3 expression is rather misleading, as it's due to a subtle shift between the two gradients shown in fig 1D, which is normal and not surprising. I may be missing the point, but I don't see the purpose of repeating the blot in 1D with a different loading order.

4) Images: please consider including zooms, and magenta rather than red.

Reviewer #1 (Comments to the Authors (Required)):

The manuscript by Jayabalan et al. reports on the crucial role of the ADP-ribose polymerase PARP10 for regulation of stress granules (SGs). The authors first investigated whether overexpression of the mono-ADP-ribose (MAR) hydrolase nsP3 or the poly-ADP-ribose (PAR) hydrolase PARG have an impact on SG number when overexpressed in U2OS cells. They found a greater reduction in SGs upon expression of the MAR hydrolase compared to the PAR hydrolase, implicating MAR removal as an important step in SG dynamics. Next, they screened 13 MAR-adding enzymes from the PARP family (using siRNA-mediated knockdown) for a potential effect on SG number, which identified PARP10 as major and PARP14 as additional hit. They further followed up the PARP10 finding. Using live cell imaging, they confirmed that shRNA-mediated knockdown of PARP10 causes a delay in the formation of G3BP1-positive granules, however GFP-tagged PARP10 does not localize to SGs, but forms distinct cytoplasmic granules. In G3BP1/2 double-KO cells, SG formation can be rescued by overexpressing GFP-G3BP1 (or a synthetic mimic), which is dependent on PARP10. The authors then seek to demonstrate PARP10-dependent MAR/PARYlation of G3BP1 (or the synthetic G3BP1 mimic), and that PARP10 knockdown alters SG composition and reduces phospho-eIF2alpha phosphorylation. They propose that MARYlation of G3BP1 (or other SG proteins) by PARP10 is a key first step in SG initiation, followed by PARYlation by other PARPs, e.g. PARP5a or PARP1. The polynucleotide-like polymer then may recruit further proteins and help in the assembly of SGs. The study nicely complements prior work implicating MAR/PAR are crucial regulators of SG dynamics and highlights a so far unknown role of PARP10 in this process. It is therefore of interest to a wide cell biology/biochemistry readership. However, the study has several technical shortcomings and could benefit from some text/figure revisions to make it more understandable and technically sound.

We thank the reviewer for the constructive feedback and for noting that the paper is “*of interest to a wide cell biology/biochemistry audience*”. We have revised the text and adjusted the figures in accordance with the suggestions. Please see our detailed response below.

Major points:

1. Tone down overstatement: On page 4, when describing their results, the authors several times mention that "MARYlation or nsP3 expression inhibits SG assembly" or that PARP10/14 knockdown results in "reduced SG assembly". This should be rephrased, as they simply demonstrate reduced SG numbers upon these treatments, however at this point it cannot be claimed that the reduced SG number is due to impaired SG assembly - it might as well be caused by accelerated SG disassembly or enhanced SG degradation. Subsequent experiments (Fig. 2/3) hint at this being indeed correct, but to not oversell their data, the reduced steady state number of SGs should be interpreted with more caution/neutrality.

We have now rephrased the data by the number of cells with SGs upon treatment when describing the result. We also add a statement “*Endpoint-based assays, which only offer a static view of SG presence in cells, may not fully distinguish whether the reduced percentage of cells with SGs observed upon PARP10 knockdown is due to impaired SG assembly, accelerated disassembly, or enhanced degradation*”

2. Figure 2I: The GFP-G3BP1 signal in the bottom row (shPARP10 line) looks much dimmer than in the upper row (WT line) - if this is a true phenotype, this is a severe concern, because SG formation is influenced by the G3BP1 concentration. So a cell line with lower levels of GFP-G3BP1 would show fewer SGs simply due to the lower protein levels and then the effect might not come from the PARP10 knockdown. To clarify this concern, the authors should demonstrate that both cell lines have equal

levels of GFP-G3BP1 and if so, show more suitable snapshot images where both cell lines have (roughly) equal levels of GFP-G3BP1 and hence can be truly compared.

Thank you for highlighting this discrepancy. It was due to a technical issue during postprocessing. We have replaced the snapshot and additionally included a live-cell imaging video (*Movie 1*) corresponding to the snapshots, indicating comparable G3BP1 expression levels. Furthermore, please refer to *Fig. 4E*, which indicate little differences in GFP-G3BP1 expression levels between the WT and shPARP10 cell lines.

3. Methods details on how the GFP-ki-G3BP1 line and stable / dox-inducible GFP-PARP10-lines were generated are missing completely. This should be included in the methods sections, and experimental details (e.g. selection media, dox concentration) should be given. Similarly, methods details (qPCR) how the PARP knockdown efficiencies (shown in *Fig. S1A* and *S2B*) were determined are missing.

Thank you for bringing the missing details to our attention. The GFP-ki-G3BP1 cell line was kindly provided by Dr. Paul Taylor's group, as noted in the *Acknowledgements* section. We have now included the details regarding how we generated the shRNA knockdown cells with the GFP-ki-G3BP1 cell line as well as doxycycline selection and the qPCR methods in the *Methods* section.

4. *Fig. 2C*: As GFP-tagging often alters a protein's condensation/localization behavior, the authors should examine PARP10 fused to a small epitope tag (ideally fused to the N- or C-terminus to exclude tagging artifacts), to validate the (punctate, but not SG-associated) localization seen for the GFP-tagged construct.

We would like to point out that the presence of PARP10 in condensates has been consistently observed by multiple independent groups using various tags and tagging at different termini of the protein. We have now added a statement to clarify this point: "*Cells expressing PARP10 formed condensates distinct from SGs (Fig. S2A)³⁵, in agreement with previous studies showing that PARP10 is capable of forming condensates, regardless of the type or position of the protein tag^{36,37}.*"

5. *Figure 2D/E*: A non-PARP10 overexpressing parental control is missing in both panels (particularly relevant would be to show in *Fig. 2E* what % of untransfected cells had SGs). Another major concern is that the catalytically inactive mutant (G888W) is expressed at much lower levels than the WT, hence the observed effect (fewer SGs in mutant-expressing line) could also be due to the lower expression levels. Hence, the authors should repeat the experiment ensuring equal expression of WT and G888W mutant (if necessary by adjusting the DOX concentration to achieve equal expression). If it is not possible to achieve this, the experiments cannot be properly interpreted and should be removed from the manuscript.

Thank you for your valuable point. We used the condition of 0.2 mM arsenite for 30 minutes, under which ~90% of untransfected cells formed SGs. This can be identified by examining neighboring cells as well as the full-field image (please see the raw data file below). Furthermore, we were unable to achieve G888W expression at WT levels, even after doubling the transfection concentration of the plasmid encoding G888W, suggesting that its catalytic activity may be critical for proper expression and/or function of PARP10 or that the high-level expression of the catalytic dead mutant is somehow suppressed by an unknown mechanism. Yet, even with low expression, the G888W mutant is sufficient to reduce the number of SGs. Therefore, we propose that catalytic activity is necessary for SG assembly, consistent with the observation that PARP10 KD impairs SG assembly (*Fig. 2*) and reduces G3BP1 ADP-ribosylation (*Fig. 3*).

[Figure removed by editorial staff per authors' request]

6. Figure 3E/F: GFP-IP should be performed under denaturing conditions (e.g. using 1% SDS in the lysis buffer + heating, followed by dilution into a native buffer that allows GFP-IP), otherwise one cannot be sure that the identified signals really come from direct modification of G3BP1 or the mimic (it could be from co-precipitated modified proteins that run at the same MW).

Thank you for your suggestion. Initially, we attempted to perform IP under denaturing conditions; however, due to the labile nature of ADP-ribose under harsh treatment (PMID: 36368907), we were unable to detect a clear ADP-ribosylation signal under these conditions.

[Figure removed by editorial staff per authors' request]

Fig. R1: (a) U2OS cells were lysed, immunoprecipitated with G3BP1 antibodies and incubated with buffer, 2 μ M purified PARP10 catalytic domain (PARP10 cd). After 1 h incubation, samples were subjected to western blotting against ADPr-reagents. (b) G3BP1/2 dKO cells were transfected with either GFP-vector, GFP-tagged G3BP1 WT or mimic constructs for 24 h. Post transfection, cells were immunoprecipitated using anti-GFP antibodies, and blotted against pan-ADPr reagent.

7. Figure 3E/F: The G3BP1-mimic has a strikingly different band pattern in the PAR blot (single band) compared to the high MW smear seen for G3BP1 WT, yet the authors do not comment on where this difference (likely) comes from. Is the G3BP1-mimic MARylated instead of PARylated (have they reprobed the mimic-IP with a MAR-specific antibody?) The authors should make an effort to clarify

what the difference in ADP-ribosylation pattern between the WT and mimic is, or at least point out the striking difference and speculate what the molecular difference could be. If the G3BP1-mimic is indeed MARylated instead of PARylated, this would be an interesting finding, hence this possibility should be explored and either validated or disproved.

We thank the reviewer for this intriguing question. Figures 3E/F represent our earlier observations, and antibodies specific for MAR and PAR only became commercially available recently. We have now repeated the experiment and probed with both MAR- and PAR-specific antibodies. We found that both G3BP1 WT and the mimic are in MARylated and PARylated form. This new data has been included in the revised manuscript (Fig. 4B).

Minor points:

- Page 3, line 37: mRNAs are never truly "naked" in the cell (still RBP-bound), rephrase.

We have revised the wording of this sentence as follows, *"These mRNAs engage in RNA-RNA interactions, condensing with one another, as well as with translation factors and RNA-binding proteins in the cytoplasm to form SGs"*

- Page 3, line 40/41: Last sentence ("Yet,...) is very generic and in my view just disrupts the flow, so I recommend to delete it.

We removed the word "Yet" from the sentence.

- Page 3, line 59: in addition to ref 24, ref 31 and 39 would be relevant and fitting citations here.

We have now added both references.

- Page 3, line 48: When talking about PARPs and PARG as known SG components, the authors could check whether these factors are also found in published SG proteomes / proximity maps (e.g. Jain et al., Cell 2015; Youn et al., Mol Cell 2018) and potentially mention this here (PMID: 26777405; PMID: 29395067).

Thank you for your suggestion. Indeed, both reports identified PARP12, with one paper also identified PARG. We have now included the following statement: *"Proteomics analyses have further confirmed PARP12 and PARG as components in biochemically purified SGs^{22,23}."*

- Page 4, lines 79: The authors state that PARG and nsP3 are co-expressed "at different expression levels". This statement is misleading, as the IF images in Fig. 1A show roughly equal expression levels across the titration. Presumably, as the DNA is titrated up, simply the % of transfected cells increases (hence the more prominent band in the Western blot for higher DNA conc.), however on the single cell level the expression level is likely constant. In line with this, there is not really a dose-dependent effect observed (quantification in Fig. 1B). To not mislead the reader, the statement about the "different expression levels" should be rephrased, or the titration could be omitted altogether and just 1 DNA concentration could be shown...

Thank you for your suggestion. We revised the statement from "expression" to "concentration" as follows:

"To further investigate the effect of MAR vs PAR hydrolases on SG assembly, we co-expressed PARG and nsP3, varying the DNA construct concentration of one while keeping the other constant. First, we

maintained the amount of plasmid encoding the PAR-degrading enzyme PARG constant while gradually increasing the one for MAR-degrading nsP3."

- Page 4, lines 95-97: Check grammar, sounds like a verb is missing here...

Thank you for highlighting the mistake. We have revised the sentence accordingly.

- Figure 1 would be more readable if the text and figure order were aligned (titration of flag-nsP3 described first in the text, but shown second in Fig. 1A and B - it would be more readable if they showed the nsP3 titration/quantification panel first, and the GFP-PARG titration data second).

We have updated the figure panels accordingly to the text.

- Similarly, Fig. 3 would be more readable/understandable if the G3BP1 mimic (shown throughout all panels) would be mentioned from the beginning of this results paragraph, along with G3BP1 WT, as it is quite confusing to always see this mimic in all figure panels, but not know why it's there or what it is. This confusion could be avoided by rewriting the results text and by introducing the mimic together with G3BP1 WT (on p. 7 already). Moreover, it should be pointed out why this mimic has been used and what can be learnt from it (not entirely clear to me because too little info on the mimic is given).

We have revised the flow accordingly as follows:

"G3BP1 and its paralog G3BP2 are jointly essential SG components, as G3BP1/2 double knockout (dKO) cells fail to form SGs upon various stress conditions, including arsenite treatment^{27,29}. However, reintroducing at least one of the G3BPs (either G3BP1 or G3BP2) restores SG formation in the dKO²⁹. A recent study further indicates that SG formation in G3BP1/2 dKO cells can also be restored by a synthetic construct mimicking G3BP1²⁹. This mimic replaces G3BP1's essential domains responsible for SG assembly with analogous domains from other proteins (Fig. 3A)."

- Figure 1E, are the fraction numbers mislabelled? In figure 1D, fraction no.1 has no ADPr signal, however in 1E there is a strong ADPr signal in fraction no. 1. Correct or explain the discrepancy.

We thank the reviewer for noticing the wrong labeling. As it is a repeat of Fig. 1D, as per reviewer 2's suggestion, we have removed Fig. 1E.

- Figure 1E: Since no quantification or statistics are shown: Authors should state in the legend how often this experiment was performed and that the shown WB is representative of n= ? experiments.

Thank you for pointing out the missing details. We have now included the number of repeats performed in figure legends.

- Fig. 2D labelling is hard to understand, it took me a long time to find out what they mean with "WT" ◊ better label as "CONST. WT" (as opposed to DOX-inducible WT or mutant...)

Thank you for the suggestion. We have updated the labels accordingly.

- For live cell imaging (methods): mention the number of cells that were imaged/analyzed.

We have now included the number of cells analyzed in figure legends.

Reviewer #2 (Comments to the Authors (Required)):

Re: Jayabalan et al., "PARP10 is critical for stress granule initiation"

This is an investigation of the influence of ADP-ribosylation on formation of stress granules (SGs). The enzymes adding ADP-ribosyl units to target proteins are called poly-ADP-ribosyltransferases (PARPs). Despite their name only some of these add multiple ADP-ribosyl monomers to the target protein; others add only one subunit (mono-ADP-ribosylation; MARYlation).

Here, the authors found that knockdown of PARP10, and to a lesser degree PARP14, results in a reduction and delay of SG formation. No other PARP knockdown caused a noticeable defect. The well-studied SG component G3BP1 was established as one of the substrates for PARP10. Based on previous comparisons between the effects on SG formation by a viral poly-ADP-ribosyl degrading enzyme and its cellular counterpart and experiments in the current manuscript, the authors inferred that addition of the first ADP-ribosyl unit (MARYlation) is critical for SG assembly. The authors further demonstrate that G3BP1 is mostly PARYlated, leading to a model where this protein is first MARYlated by PARP10, and then poly-ADP-ribosylated, building on that first ADP-ribosyl unit by other PARPs.

Overall, the manuscript is of high technical quality and the claims are supported by experimental data. It is of interest to a wide scientific community. There are however some questions that should be addressed, see below.

We appreciate the reviewer for recognizing “the high technical quality of our manuscript and its relevance to a wide scientific community”, as well as for providing constructive criticism. Please see our detailed response below.

- One obvious question that could have been answered within the scope of this manuscript is what happens with a double knockdown of PARP10 and PARP14. Given the authors' working model, where addition of the first ADP ribosyl subunit (MARYlation) is a starting point for SG assembly, this would be suspected to result in a strong effect. This would be an interesting addition to the paper, and should not be too demanding to perform. The implications of a PARP10/PARP14 double knockdown could be particularly interesting for the model of SG formation since PARP14 is an SG component but PARP10 is not (line 256). Is it possible that ADP-ribosylation both within and outside of SGs are required for SG assembly?

Thank you for raising this interesting possibility. Indeed, we hypothesized that a double knockdown of PARP10 and PARP14 might have a synergetic effect on SG inhibition. However, we did not observe a stronger effect. Instead, knocking down both PARP10 and PARP14 resulted in a similar number of SGs as observed with PARP10 knockdown alone. One possible explanation is that these two PARPs may have redundant functions or present in overlapping pathway. Given the stronger effect of PARP10 KD alone on SG inhibition, we chose to focus our current study on further investigating its role in SG assembly. While we cannot rule out the possibility that ADP-ribosylation is required both within and outside of SGs, this warrants further investigation. We have now included these data in the main manuscript as Fig. S1B-C.

- The authors used arsenite to induce SGs. The protein composition of SGs is affected by the specific

stress used to induce them. Is it possible that the requirement of ADP ribosylation for SG assembly is also affected by the type of stress?

This is an insightful question regarding whether the requirement for ADP-ribosylation in SG assembly varies depending on the type of stress. In this study, we examined SG formation under multiple stress conditions. We used sodium arsenite, which activates HRI kinase and induces SGs via eIF2 α phosphorylation, and also G3BP1 overexpression, which activates PKR kinase and similarly promotes eIF2 α -dependent SG formation. Additionally, we now tested SG induction with pateamine A, which bypasses eIF2 α phosphorylation, and observed that PARP10 knockdown did not impair SG formation in this context (Fig. 2SC-D). These results suggest that PARP10-mediated regulation of SGs is specific to eIF2 α -dependent stress responses.

In our prior work, we also demonstrated that ADP-ribosylation is critical for SG formation during alphavirus infection (PMID: 33547245), a condition that also involves eIF2 α phosphorylation. It is plausible that the requirement for ADP-ribosylation—and the specific substrates involved—varies with different types of stress, especially given that SG composition can differ depending on the inducing stimulus (PMID: 30100264; PMID: 30728452). However, the precise mechanisms by which ADP-ribosylation influence SG assembly in a stress-specific manner, and whether PARP10 or other PARPs are differentially involved, remain to be determined.

- Another question is this: if PARP10 has such a fundamental role in SG assembly, how come it has escaped detection in previous screens, genome-wide or directed, for factors required for SG formation? There have been a number of these over the years, e.g. PMID 18794846, PMID 32393832. Was PARP10 for some reason not included in previous screens? Or is the effect on SG formation too limited to be reliably detected in a high-throughput format?

A potential explanation for this discrepancy is the redundancy among PARP family members in different cellular contexts, which we observed between PARP10 and PARP14 in our study. It is possible that MARylation is essential for SG assembly but is not necessarily dependent on the activity of a single PARP. Supporting this notion, a recent study (PMID: 39760726) demonstrated that MARylation by PARP14 plays a critical role in regulating SG assembly in ovarian cancer cells. Thus, the contribution of PARP10 may vary depending on the cellular context or may be masked by compensatory activity from other PARPs. We have elaborated on these points in the “Limitations” section of the manuscript.

“Our data point to PARP10 as the primary PARP involved in SG assembly. However, it is possible that different cell lines might depend on other MAR-adding enzymes, such as PARP14.”

Minor remarks:

- The authors constructed a synthetic protein designed to mimic G3BP1 on an overall structural level despite having no primary sequence similarity, and showed that it also functions as a target of PARP10. However, we are not offered much in terms of a rationale why this experiment was performed in the first place, nor how its outcome should be interpreted. The authors claim that "This finding raises the intriguing possibility that ADP-ribosylation of any core SG component could be sufficient to initiate SG assembly, irrespective of its identity", but there is no evidence to support that idea.

We apologize for being unclear about our rationale. The current understanding in the SG field is that G3BP1/2 are the core proteins required for SG formation under most stress conditions. Recently, Dr.

Paul Taylor's group reported that a G3BP1-like synthetic construct is sufficient to form SGs when added back to G3BP1/2 knockout cells. In this manuscript, we identify PARP10 as an upstream factor necessary for SG formation. To demonstrate the importance of PARP10's role, we tested both G3BP1 and its functional mimics and found that both require PARP10 for SG assembly. We have now introduced this concept earlier: *"G3BP1 and its paralog G3BP2 are jointly essential SG components, as G3BP1/2 double knockout (dKO) cells fail to form SGs upon various stress conditions, including arsenite treatment^{26,28}. However, reintroducing at least one of the G3BPs (either G3BP1 or G3BP2) restores SG formation in the dKO²⁸. A recent study further indicates that SG formation in G3BP1/2 dKO cells can also be restored by a synthetic construct mimicking G3BP1²⁸. This mimic replaces G3BP1's essential domains responsible for SG assembly with analogous domains from other proteins (Fig. 3A)."*

We have also explained the implications of the experiment more clearly: *"A surprising outcome of our work is that the synthetic G3BP1 mimic, which replaces the NTF2-like, RRM, and IDR modules with analogous domains from unrelated proteins²⁸ nevertheless requires PARP10. This observation suggests that PARP10 targets generic physicochemical features rather than a specific G3BP1 sequence motif. It also underscores the utility of minimalist scaffolds for dissecting the rules of condensate regulation: by substituting individual domains, we can test whether MARYlation acts on RNA-binding modules, dimerization domains, or disordered tails for SG formation."*

- Line 166: "G3BP1 and its paralog G3BP2 are essential SG components" should be "...are synthetically essential..."

As "synthetically essential" may not be a common term, we have edited as follows:

"G3BP1 and its paralog G3BP2 are jointly essential SG components, as G3BP1/2 double knockout (dKO) cells fail to form SGs upon various stress conditions, including arsenite treatment^{27,29}."

Reviewer #3 (Comments to the Authors (Required)):

The submitted work focuses on the role of PARP10 in SG formation. The lab is pioneer in the subject and had previously reported compelling evidence that parylation influences SG condensation. Here it's proposed that marylation (the addition of a single ADP ribose) by PARP10 is a rate-limiting step in the initiation of SG formation.

Experiments are in general technically well performed. However I don't fully agree with a number of rationales and interpretation, as follows.

We thank the reviewer for stating that the manuscript is *"technically well performed"* and their constructive criticisms.

1. A major concern relates to the potential pleiotropic effects of PARP10 loss-of-function. When reading the MS, from the very beginning, I felt that an important issue was missing, and is whether stress sensing was affected by PARP10 KD. Marylation is a constitutive post-translational modification and a yet unknown number of proteins can potentially be affected. In addition, as authors explained, ADP-ribosylation is connected to NAD metabolism, and the effect of PARP10 KD in the overall cellular fitness is unknown.

Acute stress sensing is greatly influenced by cell health. Preconditioning by previous stress insults, either chronic or acute, may reduce the stress response, with lower eIF2alpha phosphorylation and lower SG formation in several examples.

Thus, a main question is whether the strong reduction in the phosphorylation of eIF2 α (down to 50 % in figure 4D) is the main cause of defective SG formation upon PARP10 KD.

A) A potentially useful experiment to solve this important issue is to attempt to induce SGs that form independently of eIF2 α phosphorylation. Mitochondrial inhibitors may work. Also, translation initiation inhibitors such as hippuristanol were shown to induce SGs independent of eIF2 α phosphorylation.

We thank the reviewer for their insightful comments regarding the potential pleiotropic effects of PARP10 knockdown and the specificity of its impact on SG formation. We agree that it is important to determine whether the observed phenotype is due to general cellular dysfunction or specifically linked to certain SG formation pathways. To address this concern, we conducted additional experiments using pateamine A, a translation initiation inhibitor known to induce SG formation independently of eIF2 α phosphorylation. Notably, we observed no significant effect on SG formation upon PARP10 knockdown under these conditions (Fig. S2C-D). These findings suggest that the defect in SG formation by PARP10 knockdown is not due to global cellular impairment or general translational inhibition, but rather is specific to stress conditions that involve eIF2 α phosphorylation.

B) In the same line, is the arsenite-induced translational silencing affected by PARP10 KD? (this can be easily assessed by puromycylation)

We thank the reviewer for this insightful suggestion. Given that PARP10 knockdown consistently reduces eIF2 α phosphorylation, we agree that one would expect increased translation during arsenite-induced stress. To test this, we performed puromycylation assays in five independent experiments. While the reduction in p-eIF2 α levels was consistently observed, the puromycylation results were variable

These inconsistent findings suggest that the relief of translational repression via reduced eIF2 α phosphorylation may not fully account for the observed defects in stress granule formation upon PARP10 knockdown. This raises the possibility that PARP10 may regulate SG dynamics through additional mechanisms beyond translation initiation — for example, by modulating ADP-ribosylation of core SG components such as G3BP1 and compositional alteration in translation factors within SGs.

[Figure removed by editorial staff per authors' request]

C) Levels of arsenite-induced phospho 2alpha should be controlled upon PARG and nsP3 transfection, to allow the conclusion that the effect is downstream of stress sensing. (I don't recommend to assess translation by puromycilation in these cases, as overexpression of transfected proteins seriously interfere with the assay)

We thank the reviewer for this important point. To determine whether the observed effects of PARG and nsP3 on SG formation occur downstream of stress sensing, we assessed the levels of arsenite-induced phosphorylated eIF2 α following transfection of PARG or nsP3. As shown in Fig. 1D, neither manipulation affected the induction of p-eIF2 α in response to arsenite, indicating that the upstream stress sensing pathway remains intact. This result supports the conclusion that the effects of PARG and nsP3 on stress granule formation occur downstream of eIF2 α phosphorylation, likely through interference with the ADP-ribosylation of key SG components.

2. Figure 3E, F. Both G3BP-GFP and G3BPmimic-GFP are parylated by PAR10. Was this expected? Are the amino acids involved known in each case? Should we expect that any overexpressed SG protein will be parylated? Would it be possible to include a control of a transfected protein that won't be parylated?

It was expected that G3BP1-GFP would be ADP-ribosylated; however, the specific enzymes responsible for this modification remain unclear. Notably, the observed smear in the ADP-ribosylation pattern, which is indicative of PARylation, made it unexpected that PARP10, a PAR-adding enzyme, would be involved in this process.

For the G3BP mimic, we did not initially anticipate that it would undergo ADP-ribosylation. The purpose of this experiment was to determine whether a protein with an equivalent modular domain architecture to G3BP1 could functionally substitute for it. Interestingly, the observation that G3BP mimic-GFP is ADP-ribosylated, and that this modification is dependent on PARP10, suggests that the G3BP mimic can indeed replace G3BP1 functionally, including in the context of ADP-ribosylation.

We do not expect that all overexpressed SG proteins will be ADP-ribosylated. In our previous work (PMID: 21596313; PMID: 33547245), we demonstrated that SG components, including GFP-PABP, GFP-eIF3g, and GFP-eIF3i, are not ADP-ribosylated upon transfection. Additionally, no PARylation was observed in the transfected GFP control used in our experiments.

3. Figure 4B. PARP10 KD reduces SG condensation, so that less G3BP-GFP is expected to be recovered in the pellet. I understand that the loading was adjusted to have equal amounts of G3BP-GFP (first row). Simultaneously, several initiation factors normally present in SGs are less abundant relative to G3BP. How is this interpreted? G3BP condensates contain less abortive translation initiation complexes? Is this indicative of defective translational silencing (see above)?

We thank the reviewer for the insightful question. In our previous work, we showed that certain translation initiation factors, such as eIF3g, can bind ADP-ribose, and that alphaviruses disrupt ADP-ribosylation to disassemble SGs and release translation machinery [PMID: 33547245]. In this study, we identify PARP10 as a key regulator of G3BP1-mediated stress granule assembly.

In Figure 4E, although similar amounts of GFP-G3BP1 (a core SG scaffold protein) were recovered in the SG-enriched pellet fractions, several translation initiation factors were less abundant upon PARP10 knockdown. This suggests that ADP-ribosylation mediated by PARP10 is important not only for SG condensation but also for the recruitment or retention of abortive translation initiation complexes within SGs.

Rather than reflecting enhanced translation per se, this observation supports the idea that SGs formed in the absence of PARP10 are compositionally deficient—particularly in translation-related components—potentially due to altered protein-protein or protein-RNA interactions required for their incorporation. These findings further underscore the role of ADP-ribosylation in shaping SG composition and may help explain the impaired SG maturation observed upon PARP10 knockdown.

In summary, the model proposed in Fig4E (as well as the title and abstract) largely overstates the relevance of G3BP phosphorylation in driving SG formation. Instead, the upstream defective eIF2 α phosphorylation may fully explain the effect. If the target residue(s) in G3BP can be identified, a point mutation that abrogates phosphorylation might directly assess the role of G3BP phosphorylation.

We appreciate the reviewer's thoughtful critique regarding the model presented in Fig. 4F. We agree that reduced eIF2 α phosphorylation upon PARP10 knockdown partly contributes to the observed defect in stress granule formation, and we have now revised the abstract and incorporated these comments in our discussion.

“Regardless, this study emphasizes the crucial role of ADP-ribosylation mediated by PARP10, a MAR-adding enzyme, to maintain the phosphorylation of the translation factor eIF2 α and the presence of other translation factors within the SG core. These data suggest that PARP10 may mediate SG assembly at multiple steps, from modulation of stress signaling to stress granule condensation and composition.”

However, we would like to highlight that our data also indicate that PARP10 regulates SG composition beyond simply modulating translation initiation. Specifically, we show that several translation initiation factors are less efficiently incorporated into SG-enriched fractions despite similar recovery of G3BP1 (Fig. 4E). These findings suggest that PARP10-mediated ADP-ribosylation may influence the biophysical properties of G3BP1 or its ability to recruit interacting partners during SG assembly.

We fully agree that identifying the specific ADP-ribosylation site(s) on G3BP1 and testing their functional relevance via site-directed mutagenesis would provide a more definitive test of this hypothesis. We are actively pursuing this direction, and plan to incorporate these experiments in future studies.

MINOR:

1). Figure 1D (the effect of MAR-degrading nsP3) and line 98. " a reduction (of PAR signal) in fractions enriched with SG-associated proteins". It appears that all lanes with significant OD (from 2 to 6) have reduced ADPribose signal. (Should we expect an effect specific of SG components?)

Indeed, there are several SG components that could be ADP-ribosylated as previously reported (PMID: 30100264; PMID: 30728452; PMID: 39760726) and those modification might be reduced by MAR-degrading nsP3. In our current study, we chose to characterize the core SG component G3BP1 (Figs. 3 and 4) since we have previously reported that G3BP1 is ADP-ribosylated during stress (PMID: 21596313) and viral infection (PMID: 33547245). Our future aim is to perform proteomic analysis of ADP-ribosylated SG proteome to define other modified proteins.

2) Figure 1E. The apparent change in pattern of lines 3 and 4 upon nsP3 expression is rather misleading, as it's due to a subtle shift between the two gradients shown in fig 1D, which is normal and not surprising. I may be missing the point, but I don't see the purpose of repeating the blot in 1D with a different loading order.

The original intention for Figure 1E is to demonstrate the difference side-by-side of the same fraction in cells expressing GFP vs. GFP-nsP3. However, we agreed that the data is redundant and have removed Figure 1E.

--Please note that item 3 was not included in the original reviewer's comments.--

4) Images: please consider including zooms, and magenta rather than red.

Wherever possible, we have adjusted the images. Specifically, we enlarged the following microscopic images (Fig. 1A; Fig. 2A-B, F, I; Fig. 3C; Fig. S1D, F; Fig. S2A) and changed color from red to magenta, as suggested.

August 25, 2025

Re: Life Science Alliance manuscript #LSA-2024-03026-TR

Anthony Leung
Johns Hopkins University
615 N. Wolfe Street
BSPH/E8647/Leung Lab
Baltimore, MD 21205

Dear Dr. Leung,

Thank you for submitting your revised manuscript entitled "PARP10 is Critical for Stress Granule Initiation" to Life Science Alliance. The manuscript has been seen by the original Reviewers 2 and 3 whose comments are appended below. While the reviewers continue to be overall positive about the work in terms of its suitability for Life Science Alliance, some important issues remain.

While the regulation of G3BP1 by PARP10 modification is well-supported, Reviewer 3 remains concerned over other potential roles of PARP10. This reviewer points to the claim that PARP10 is required only for eIF2alpha-dependent SG formation whereas loss of PARP10 significantly reduces eIF2alpha phosphorylation itself. This discrepancy should be better confronted in describing the results and in the discussion. In addition, and potentially related to the above point, changes to global translation measured using puromycylation upon PARP10 depletion were variable. While this result leaves a clear answer unavailable, readers may find this observation valuable. A revised manuscript should directly confront these two points, ideally including the puromycylation data. The variable/inconsistent nature of these results do not reduce the priority of this work for publication so long as they are accurately presented.

Our general policy is that papers are considered through only one revision cycle; however, given that the suggested changes are relatively minor, we are open to one additional short round of revision. Please note that I will expect to make a final decision without additional reviewer input upon re-submission.

Please submit the final revision along with a letter that includes a point by point response to the comments of Reviewer 3.

To upload the revised version of your manuscript, please log in to your account: <https://lsa.msubmit.net/cgi-bin/main.plex>
You will be guided to complete the submission of your revised manuscript and to fill in all necessary information.

-- A letter addressing the reviewer comments point by point.

B. MANUSCRIPT ORGANIZATION AND FORMATTING:

Sincerely,

Tim Fessenden
Scientific Editor

Reviewer #2 (Comments to the Authors (Required)):

In their revised version, the authors have adequately responded to my criticism, as specified below.

- One obvious question that could have been answered within the scope of this manuscript is what happens with a double knockdown of PARP10 and PARP14. Given the authors' working model, where addition of the first ADP ribosyl subunit (MARylation) is a starting point for SG assembly, this would be suspected to result in a strong effect. This would be an interesting addition to the paper, and should not be too demanding to perform. The implications of a PARP10/PARP14 double knockdown could be particularly interesting for the model of SG formation since PARP14 is an SG component but PARP10 is not (line 256). Is it possible that ADP-ribosylation both within and outside of SGs are required for SG assembly?

Thank you for raising this interesting possibility. Indeed, we hypothesized that a double knockdown of PARP10 and PARP14 might have a synergetic effect on SG inhibition. However, we did not observe a stronger effect. Instead, knocking down both PARP10 and PARP14 resulted in a similar number of SGs as observed with PARP10 knockdown alone. One possible explanation is that these two PARPs may have redundant functions or present in overlapping pathway. Given the stronger effect of PARP10 KD alone on SG inhibition, we chose to focus our current study on further investigating its role in SG assembly. While we cannot rule out the possibility that ADP-ribosylation is required both within and outside of SGs, this warrants further investigation. We have now included these data in the main manuscript as Fig. S1B-C.

The authors now present the outcome in the double knockdown of PARP10 and PARP14 in the manuscript. The double knockdown does not display a stronger effect on SG formation than of the single PARP10 KD, the stronger of the two KD. This discounts the idea of a synergistic effect on two parallel pathways, which would be expected to be stronger than either of the single KD. This is all clear and good.

However, the authors interpret this result as "redundant functions or present in overlapping pathway". "Redundant function" does not mean anything particular as it can invoke other, unspecified, players including additional PARP enzymes. "Overlapping pathways" is similarly unspecific as it says nothing about in which way the pathways would interact. Instead, a situation where the double mutant displays the same phenotype as the single mutant with the stronger effect, is a typical result from a situation where the two proteins act in the same linear pathway.

- The authors used arsenite to induce SGs. The protein composition of SGs is affected by the specific stress used to induce them. Is it possible that the requirement of ADP ribosylation for SG assembly is also affected by the type of stress?

This is an insightful question regarding whether the requirement for ADP-ribosylation in SG assembly varies depending on the type of stress. In this study, we examined SG formation under multiple stress conditions. We used sodium arsenite, which activates HRI kinase and induces SGs via eIF2cx phosphorylation, and also G3BP1 overexpression, which activates PKR kinase and similarly promotes eIF2cx-dependent SG formation. Additionally, we now tested SG induction with pateamine A, which bypasses eIF2cx phosphorylation, and observed that PARP10 knockdown did not impair SG formation in this context (Fig. 2SC-D). These results suggest that PARP10-mediated regulation of SGs is specific to eIF2cx-dependent stress responses.

In our prior work, we also demonstrated that ADP-ribosylation is critical for SG formation during alphavirus infection (PMID: 33547245), a condition that also involves eIF2cx phosphorylation. It is plausible that the requirement for ADP-ribosylation-and the specific substrates involved-varies with different types of stress, especially given that SG composition can differ depending on the inducing stimulus (PMID: 30100264; PMID: 30728452). However, the precise mechanisms by which ADP-ribosylation influence SG assembly in a stress-specific manner, and whether PARP10 or other PARPs are differentially involved, remain to be determined.

These additional experiments do provide information about the generality of their observations on the importance of ADP-ribosylation for SG formation, and strengthen the manuscript.

- Another question is this: if PARP10 has such a fundamental role in SG assembly, how come it has escaped detection in previous screens, genome-wide or directed, for factors required for SG formation? There have been a number of these over the years, e.g. PMID 18794846, PMID 32393832. Was PARP10 for some reason not included in previous screens? Or is the effect on SG formation too limited to be reliably detected in a high-throughput format?

A potential explanation for this discrepancy is the redundancy among PARP family members in different cellular contexts, which we observed between PARP10 and PARP14 in our study. It is possible that MARylation is essential for SG assembly but is not necessarily dependent on the activity of a single PARP. Supporting this notion, a recent study (PMID: 39760726) demonstrated that MARylation by PARP14 plays a critical role in regulating SG assembly in ovarian cancer cells. Thus, the contribution of PARP10 may vary depending on the cellular context or may be masked by compensatory activity from other PARPs. We have

elaborated on these points in the "Limitations" section of the manuscript.

"Our data point to PARP10 as the primary PARP involved in SG assembly. However, it is possible that different cell lines might depend on other MAR-adding enzymes, such as PARP14."

Yes, it is plausible that the differing screening conditions may have masked the effect of PARP10 knockdown in other publications.

Minor remarks: - The authors constructed a synthetic protein designed to mimic G3BP1 on an overall structural level despite having no primary sequence similarity, and showed that it also functions as a target of PARP10. However, we are not offered much in terms of a rationale why this experiment was performed in the first place, nor how its outcome should be interpreted. The authors claim that "This finding raises the intriguing possibility that ADP-ribosylation of any core SG component could be sufficient to initiate SG assembly, irrespective of its identity", but there is no evidence to support that idea.

We apologize for being unclear about our rationale. The current understanding in the SG field is that G3BP1/2 are the core proteins required for SG formation under most stress conditions. Recently, Dr. Paul Taylor's group reported that a G3BP1-like synthetic construct is sufficient to form SGs when added back to G3BP1/2 knockout cells. In this manuscript, we identify PARP10 as an upstream factor necessary for SG formation. To demonstrate the importance of PARP10's role, we tested both G3BP1 and its functional mimics and found that both require PARP10 for SG assembly. We have now introduced this concept earlier: "G3BP1 and its paralog G3BP2 are jointly essential SG components, as G3BP1/2 double knockout (dKO) cells fail to form SGs upon various stress conditions, including arsenite treatment^{26,28}. However, reintroducing at least one of the G3BPs (either G3BP1 or G3BP2) restores SG formation in the dKO²⁸. A recent study further indicates that SG formation in G3BP1/2 dKO cells can also be restored by a synthetic construct mimicking G3BP1²⁸. This mimic replaces G3BP1's essential domains responsible for SG assembly with analogous domains from other proteins (Fig. 3A)."

We have also explained the implications of the experiment more clearly: "A surprising outcome of our work is that the synthetic G3BP1 mimic, which replaces the NTF2-like, RRM, and IDR modules with analogous domains from unrelated proteins²⁸ nevertheless requires PARP10. This observation suggests that PARP10 targets generic physicochemical features rather than a specific G3BP1 sequence motif. It also underscores the utility of minimalist scaffolds for dissecting the rules of condensate regulation: by substituting individual domains, we can test whether MARYlation acts on RNA-binding modules, dimerization domains, or disordered tails for SG formation."

Thank you for this thorough explanation.

- Line 166: "G3BP1 and its paralog G3BP2 are essential SG components" should be "...are synthetically essential..."
As "synthetically essential" may not be a common term, we have edited as follows: "G3BP1 and its paralog G3BP2 are jointly essential SG components, as G3BP1/2 double knockout (dKO) cells fail to form SGs upon various stress conditions, including arsenite treatment^{27,29}."

As long as it gets clear that G3BP1 and G3BP2 are not essential for this function on their own, this formulation is fine.

Reviewer #3 (Comments to the Authors (Required)):

I found the work much improved. However, a few issues still need attention.

An important issue is the conceptual framework. The work clearly shows an effect of PAR10 on the initial stages of SG formation, which is an important novelty, but fail to identify the mechanism. Several interpretations and the model are strongly biased to a role of MARYlation in the control of aggregation by G3BP and others -for which there is no clear evidence-, and minimize an important biological effect downstream of PARP10 loss-of function, that is defective 2alpha phosphorylation -which is the triggering event for arsenite-induced SGs-, as the new pateamine experiment suggests.

PARPs in general modifies proteins, DNA and RNA (ref # 40), which is a key SG component. In addition, the ER-resident 2alpha kinase PERK (which is activated by arsenite) is MARYlated to gain full activation, and this may be connected to the slower 2alpha phosphorylation described here. In addition, among many other cellular pathways, PARP10 is implicated in DNA repair and this may affect the cell cycle, which in turn influences SG formation. I'm summarizing all this just to make clear that to reduce the effect of a widespread post-translational modification (comparable to ubiquitination or sumoylation!) to a single event (G3BP MARYlation in this case) is not an easy task, unless mutants G3BP with defective ADP-ribosylation were available. Given that this tool is not currently available (as for the case of RACK, a PAR14 target, ref 34), all correlations/anticorrelations between PAR10 activity and SG formation should be interpreted with extreme caution and more importantly, correlations cannot be taken as proof of a cause-consequence relationship.

In other words, the effect of PAR10 on the first stages of SG formation is clear, but the MS is biased when proposing a mechanism.

I'm indicating below previous points that remain unsolved or need further attention

PREVIOUS 1. A major concern relates to the potential pleiotropic effects of PARP10 loss-of-function. When reading the MS, from the very beginning, I felt that an important issue was missing, and is whether stress sensing was affected by PARP10 KD. Marylation is a constitutive post-translational modification and a yet unknown number of proteins can potentially be affected. In addition, as authors explained, ADP-ribosylation is connected to NAD metabolism, and the effect of PARP10 KD in the overall cellular fitness is unknown.

Acute stress sensing is greatly influenced by cell health. Preconditioning by previous stress insults, either chronic or acute, may reduce the stress response, with lower eIF2 α phosphorylation and lower SG formation in several examples.

Thus, a main question is whether the strong reduction in the phosphorylation of eIF2 α (down to 50 % in figure 4D) is the main cause of defective SG formation upon PARP10 KD.

A) A potentially useful experiment to solve this important issue is to attempt to induce SGs that form independently of eIF2 α phosphorylation. Mitochondrial inhibitors may work. Also, translation initiation inhibitors such as hippuristanol were shown to induce SGs independent of eIF2 α phosphorylation.

RESPONSE: We thank the reviewer for their insightful comments regarding the potential pleiotropic effects of PARP10 knockdown and the specificity of its impact on SG formation. We agree that it is important to determine whether the observed phenotype is due to general cellular dysfunction or specifically linked to certain SG formation pathways. To address this concern, we conducted additional experiments using pateamine A, a translation initiation inhibitor known to induce SG formation independently of eIF2 α phosphorylation. Notably, we observed no significant effect on SG formation upon PARP10 knockdown under these conditions (Fig. S2C-D). These findings suggest that the defect in SG formation by PARP10 knockdown is not due to global cellular impairment or general translational inhibition, but rather is specific to stress conditions that involve eIF2 α phosphorylation.

SECOND REVISION -The pateamine experiment is crucial and the result does not support the proposed model. With this new experiment, authors find that PARP10 KD affects SG formation ONLY when 2 α phosphorylation is implicated. However, authors conclude that defective 2 α phosphorylation (down to 50%) is not the main cause of defective SG formation and seems to insist that defective G3BP marylation drives the effect.

In other words, this result reveals the occurrence of PARP10-dependent and PARP-10 independent SGs. Both of them contain G3BP (Fig S2D), so that defective G3BP marylation is unlikely to be the key. Rather, PARP-10 independent SGs are also Phospho-2 α independent, directly connecting PARP10 with 2 α Phosphorylation, in accordance with the slower Phosphorylation shown in figure 4C and 4D.

I strongly suggest to revise line 202 "suggesting that PARP10 is likely involved only in eIF2 α -dependent SG formation", to clearly state that the lack of effect in pateamine-induced SGs implicates 2 α phosphorylation downstream of PARP10 KD.

PREVIOUS -B) In the same line, is the arsenite-induced translational silencing affected by PARP10 KD? (this can be easily assessed by puromycilation)

RESPONSE. We thank the reviewer for this insightful suggestion. Given that PARP10 knockdown consistently reduces eIF2 α phosphorylation, we agree that one would expect increased translation during arsenite-induced stress. To test this, we performed puromycylation assays in five independent experiments. While the reduction in p-eIF2 α levels was consistently observed, the puromycylation results were variable - in two experiments, translation was slightly increased following PARP10 knockdown, as expected (see R3); in two others, translation was reduced; and in one, it remained unchanged.

These inconsistent findings suggest that the relief of translational repression via reduced eIF2 α phosphorylation may not fully account for the observed defects in stress granule formation upon PARP10 knockdown. This raises the possibility that PARP10 may regulate SG dynamics through additional mechanisms beyond translation initiation - for example, by modulating ADP-ribosylation of core SG components such as G3BP1 and compositional alteration in translation factors within SGs.

SECOND REVISION-This is not acceptable. Phosphorylation of 2 α implicates translation initiation inhibition and reduced overall translation. The technical limitations that affects reproducibility of this simple puromycilation assay should be solved. Authors should find the way to answer whether PARP10 KD affects translation or not, as it should be expected given the strong reduction of 2 α phosphorylation. Puromycilation is a robust assay and if this tool doesn't help to address the issue, authors should attempt alternative strategies (radioactive amino-acids, others).

PREVIOUS-3. Figure 4B. PARP10 KD reduces SG condensation, so that less G3BP-GFP is expected to be recovered in the pellet. I understand that the loading was adjusted to have equal amounts of G3BP-GFP (first row). Simultaneously, several initiation factors normally present in SGs are less abundant relative to G3BP. How is this interpreted? G3BP condensates contain less abortive translation initiation complexes? Is this indicative of defective translational silencing (see above)?

RESPONSE-We thank the reviewer for the insightful question. In our previous work, we showed that certain translation initiation factors, such as eIF3g, can bind ADP-ribose, and that alphaviruses disrupt ADP-ribosylation to disassemble SGs and release translation machinery [PMID: 33547245]. In this study, we identify PARP10 as a key regulator of G3BP1-mediated stress granule assembly.

In Figure 4E, although similar amounts of GFP-G3BP1 (a core SG scaffold protein) were recovered in the SG-enriched pellet fractions, several translation initiation factors were less abundant upon PARP10 knockdown. This suggests that ADP-ribosylation mediated by PARP10 is important not only for SG condensation but also for the recruitment or retention of abortive translation initiation complexes within SGs.

Rather than reflecting enhanced translation per se, this observation supports the idea that SGs formed in the absence of PARP10 are compositionally deficient-particularly in translation-related components-potentially due to altered protein-protein or protein-RNA interactions required for their incorporation. These findings further underscore the role of ADP-ribosylation in shaping SG composition and may help explain the impaired SG maturation observed upon PARP10 knockdown.

SECOND REVISION-First, pellets were analysed at 30 min of arsenite treatment, a time-point when SG formation in PARP10 KD cells is reduced to half of the normal values. Is it correct that similar amounts of G3BP are recovered in the pellets? Is this expected? Or rather, the amount of G3BP recovered in the pellet should correlate with the strength of SG formation?

More important, I don't agree that compositional differences of these pellets are not the consequence of the delay in SG formation (of about 10-15 min, plot in figure 2H), which is likely the consequence of slower 2alpha phosphorylation (as shown in figure 4C and 4D). In other words, PARP10 KD slow down 2alpha phosphorylation and SG formation, and thus, the pellets contain less SG material, in particular lower amounts of translation initiation factors. The interpretation of this result should be revised.

PREVIOUS- In summary, the model proposed in Fig4E (as well as the title and abstract) largely overstates the relevance of G3BP maryl原因 in driving SG formation. Instead, the upstream defective eIF2a phosphorylation may fully explain the effect. If the target residue(s) in G3BP can be identified, a point mutation that abrogates maryl原因 might directly assess the role of G3BP maryl原因.

RESPONSE -We appreciate the reviewer's thoughtful critique regarding the model presented in Fig. 4F. We agree that reduced eIF2cx phosphorylation upon PARP10 knockdown partly contributes to the observed defect in stress granule formation, and we have now revised the abstract and incorporated these comments in our discussion.

"Regardless, this study emphasizes the crucial role of ADP-ribosylation mediated by PARP10, a MAR-adding enzyme, to maintain the phosphorylation of the translation factor eIF2cx and the presence of other translation factors within the SG core. These data suggest that PARP10 may mediate SG assembly at multiple steps, from modulation of stress signaling to stress granule condensation and composition."

However, we would like to highlight that our data also indicate that PARP10 regulates SG composition beyond simply modulating translation initiation. Specifically, we show that several translation initiation factors are less efficiently incorporated into SG-enriched fractions despite similar recovery of G3BP1 (Fig. 4E). These findings suggest that PARP10-mediated ADP-ribosylation may influence the biophysical properties of G3BP1 or its ability to recruit interacting partners during SG assembly.

We fully agree that identifying the specific ADP-ribosylation site(s) on G3BP1 and testing their functional relevance via site-directed mutagenesis would provide a more definitive test of this hypothesis. We are actively pursuing this direction, and plan to incorporate these experiments in future studies.

SECOND REVISION -As I said before, the interpretation of the western blots of the pellets enriched in SGs when less SGs are formed is dubious and doesn't support the conclusion. In addition, given the limitations to specifically manipulate of G3BP maryl原因/paryl原因, while leaving most of the proteome unchanged, authors are not in the position to conclude that G3BP modifications are the cause of the observed effects on SGs.

The Discussion includes a section entitled "Working Model: MARYlation as a rate-limiting step of SG assembly" dedicated to the maryl原因 of G3BP and its effect on protein-protein interactions and aggregation. The role of 2alpha phosphorylation on SG formation and the effect of PARP10 in this triggering event is surprisingly absent. Once again, I'd like to stress this is an important finding, as PARYlation is a therapeutic target and knowing the cellular consequences of its manipulation is a valuable contribution.

Likewise, in the 4F figure caption (the model) the importance of G3BP MARYlation doesn't fit with the supporting data, which rather suggest defective 2alpha phosphorylation as the main driver for the slow response observed upon PARP10 KD.

In addition, please modify the following figure titles

Figure 2 | "PARP10 is essential for SG assembly". PARP10 is not essential. The KD just slightly delays SG formation.
Figure 3 | "PARP10-mediated ADP-ribosylation is critical for SG condensation". It's not critical.

PREVIOUS- MINOR:

1). Figure 1D (the effect of MAR-degrading nsP3) and line 98. " a reduction (of PAR signal) in fractions enriched with SG-associated proteins". It appears that all lanes with significant OD (from 2 to 6) have reduced ADPribose signal. (Should we expect an effect specific of SG components?)

RESPONSE-Indeed, there are several SG components that could be ADP-ribosylated as previously reported (PMID: 30100264; PMID: 30728452; PMID: 39760726) and those modification might be reduced by MAR-degrading nsP3. In our current study, we chose to characterize the core SG component G3BP1 (Figs. 3 and 4) since we have previously reported that G3BP1 is ADP-ribosylated during stress (PMID: 21596313) and viral infection (PMID: 33547245). Our future aim is to perform proteomic analysis of ADP-ribosylated SG proteome to define other modified proteins.

SECOND REVISION- The point I made is whether authors propose that the reduced PAR signal (now Fig 1E) was limited to or more dramatic for SG components. Figure 1E doesn't support a selective effect. Rather, as one should expect, all bands in all fractions with significant OD shows reduced PAR signal. This is perhaps another example of the lack of accuracy when interpreting the observations. Authors should reconsider whether a polysome profile is a useful strategy to analyze SG-associated proteins. Please modify line 98 and the overstatement in line 107 accordingly.

Reviewer #3 (Comments to the Authors (Required)):

I found the work much improved. However, a few issues still need attention.

An important issue is the conceptual framework. The work clearly shows an effect of PAR10 on the initial stages of SG formation, which is an important novelty, but fail to identify the mechanism. Several interpretations and the model are strongly biased to a role of marylaltion in the control of aggregation by G3BP and others -for which there is no clear evidence-, and minimize an important biological effect downstream of PARP10 loss-of function, that is defective 2alpha phosphorylation - which is the triggering event for arsenite-induced SGs-, as the new pateamine experiment suggests.

PARPs in general modifies proteins, DNA and RNA (ref # 40), which is a key SG component. In addition, the ER-resident 2alpha kinase PERK (which is activated by arsenite) is marylalted to gain full activation, and this may be connected to the slower 2alpha phosphorylation described here. In addition, among many other cellular pathways, PARP10 is implicated in DNA repair and this may affect the cell cycle, which in turn influences SG formation. I'm summarizing all this just to make clear that to reduce the effect of a widespread post-translational modification (comparable to ubiquitination or sumoylation!) to a single event (G3BP marylaltion in this case) is not an easy task, unless mutants G3BP with defective ADP-rybosilation were available. Given that this tool is not currently available (as for the case of RACK, a PAR14 target, ref 34), all correlations/anticorrelations between PAR10 activity and SG formation should be interpreted with extreme caution and more importantly, correlations cannot be taken as proof of a cause-consequence relationship.

In other words, the effect of PAR10 on the first stages of SG formation is clear, but the MS is biased when proposing a mechanism.

I'm indicating below previous points that remain unsolved or need further attention

PREVIOUS 1. A major concern relates to the potential pleiotropic effects of PARP10 loss-of-function. When reading the MS, from the very beginning, I felt that an important issue was missing, and is whether stress sensing was affected by PARP10 KD. Maryltion is a constitutive post-translational modification and a yet unknown number of proteins can potentially be affected. In addition, as authors explained, ADP-ribosylation is connected to NAD metabolism, and the effect of PARP10 KD in the overall cellular fitness is unknown.

Acute stress sensing is greatly influenced by cell health. Preconditioning by previous stress insults, either chronic or acute, may reduce the stress response, with lower eIF2alfa phosphoryaltion and lower SG formation in several examples.

Thus, a main question is whether the strong reduction in the phosphorylation of eIF2a (down to 50 % in figure 4D) is the main cause of defective SG formation upon PARP10 KD.

A) A potentially useful experiment to solve this important issue is to attempt to induce SGs that form independently of eIF2a phosphoryaltion. Mitochondrial inhibitors may work. Also, translation initiation inhibitors such as hippuristanol were shown to induce SGs independent of eIF2a phosphorylation.

RESPONSE: We thank the reviewer for their insightful comments regarding the potential pleiotropic effects of PARP10 knockdown and the specificity of its impact on SG formation. We agree that it is important to determine whether the observed phenotype is due to general cellular dysfunction or specifically linked to certain SG formation pathways. To address this concern, we conducted additional experiments using pateamine A, a translation initiation inhibitor known to induce SG formation independently of eIF2 α phosphorylation. Notably, we observed no significant effect on SG formation upon PARP10 knockdown under these conditions (Fig. S2C-D). These findings suggest that the defect in SG formation by PARP10 knockdown is not due to global cellular impairment or general translational inhibition, but rather is specific to stress conditions that involve eIF2 α phosphorylation.

SECOND REVISION -The pateamine experiment is crucial and the result does not support the proposed model. With this new experiment, authors find that PARP10 KD affects SG formation ONLY when 2alpha phosphorylation is implicated. However, authors conclude that defective 2alpha phosphorylation (down to 50%) is not the main cause of defective SG formation and seems to insist that defective G3BP marylation drives the effect.

In other words, this result reveals the occurrence of PARP10-dependent and PARP-10 independent SGs. Both of them contain G3BP (Fig S2D), so that defective G3BP marylation is unlikely to be the key. Rather, PARP-10 independent SGs are also Phospho-2alpha independent, directly connecting PARP10 with 2alpha Phosphorylation, in accordance with the slower Phosphoryaltion shown in figure 4C and 4D.

I strongly suggest to revise line 202 "suggesting that PARP10 is likely involved only in eIF2 α -dependent SG formation" , to clearly state that the lack of effect in pateamine-induced SGs implicates 2alpha phosphorylation downstream of PARP10 KD.

We have revised the statement, as suggested, and emphasized that PARP10 acts upstream of eIF2 α phosphorylation (now line 215).

PREVIOUS -B) In the same line, is the arsenite-induced translational silencing affected by PARP10 KD? (this can be easily assessed by puromycilation)

RESPONSE. We thank the reviewer for this insightful suggestion. Given that PARP10 knockdown consistently reduces eIF2 α phosphorylation, we agree that one would expect increased translation during arsenite-induced stress. To test this, we performed puromycylation assays in five independent experiments. While the reduction in p-eIF2 α levels was consistently observed, the puromycylation results were variable - in two experiments, translation was slightly increased following PARP10 knockdown, as expected (see R3); in two others, translation was reduced; and in one, it remained unchanged.

These inconsistent findings suggest that the relief of translational repression via reduced eIF2 α phosphorylation may not fully account for the observed defects in stress granule formation upon PARP10 knockdown. This raises the possibility that PARP10 may regulate SG dynamics through additional mechanisms beyond translation initiation - for example, by modulating ADP-ribosylation of

core SG components such as G3BP1 and compositional alteration in translation factors within SGs.

SECOND REVISION-This is not acceptable. Phosphorylation of 2alpha implicates translation initiation inhibition and reduced overall translation. The technical limitations that affects reproducibility of this simple puromycilation assay should be solved. Authors should find the way to answer whether PARP10 KD affects translation or not, as it should be expected given the strong reduction of 2alpha phosphorylation. Puromycilation is a robust assay and if this tool doesn't help to address the issue, authors should attempt alternative strategies (radioactive amino-acids, others).

We thank the reviewer for raising this important point. While decreased phosphorylation of eIF2 α is typically expected to release translational inhibition, our puromycylation assays following PARP10 knockdown yielded variable results across replicates and no reproducible trend. The assay itself functioned as expected in controls, as arsenite consistently reduced puromycin incorporation, indicating that the variability is not due to technical failure.

We carefully considered the reviewer's suggestion to apply alternative methods (e.g., metabolic labeling with radioactive amino acids). At present, these approaches are not feasible within the scope of this study. This discrepancy may instead reflect biological complexity: PARP10 may exert both direct and indirect effects on translation, making eIF2 α phosphorylation alone an incomplete predictor of global protein synthesis. For the current manuscript, we believe it is most rigorous to avoid presenting inconsistent translation readouts. Instead, we now emphasize in the Discussion (line 279): *"Given the involvement of PARP10 in translation factor localization and their post-translational modifications, its role in translation regulation warrants further investigation."*

PREVIOUS-3. Figure 4B. PARP10 KD reduces SG condensation, so that less G3BP-GFP is expected to be recovered in the pellet. I understand that the loading was adjusted to have equal amounts of G3BP-GFP (first row). Simultaneously, several initiation factors normally present in SGs are less abundant relative to G3BP. How is this interpreted? G3BP condensates contain less abortive translation initiation complexes? Is this indicative of defective translational silencing (see above)?

RESPONSE-We thank the reviewer for the insightful question. In our previous work, we showed that certain translation initiation factors, such as eIF3g, can bind ADP-ribose, and that alphaviruses disrupt ADP-ribosylation to disassemble SGs and release translation machinery [PMID: 33547245]. In this study, we identify PARP10 as a key regulator of G3BP1-mediated stress granule assembly.

In Figure 4E, although similar amounts of GFP-G3BP1 (a core SG scaffold protein) were recovered in the SG-enriched pellet fractions, several translation initiation factors were less abundant upon PARP10 knockdown. This suggests that ADP-ribosylation mediated by PARP10 is important not only for SG condensation but also for the recruitment or retention of abortive translation initiation complexes within SGs.

Rather than reflecting enhanced translation per se, this observation supports the idea that SGs formed in the absence of PARP10 are compositionally deficient-particularly in translation-related

components-potentially due to altered protein-protein or protein-RNA interactions required for their incorporation. These findings further underscore the role of ADP-ribosylation in shaping SG composition and may help explain the impaired SG maturation observed upon PARP10 knockdown.

SECOND REVISION-First, pellets were analysed at 30 min of arsenite treatment, a time-point when SG formation in PARP10 KD cells is reduced to half of the normal values. Is it correct that similar amounts of G3BP are recovered in the pellets? Is this expected? Or rather, the amount of G3BP recovered in the pellet should correlate with the strength of SG formation?

More important, I don't agree that compositional differences of these pellets are not the consequence of the delay in SG formation (of about 10-15 min, plot in figure 2H), which is likely the consequence of slower 2alpha phosphorylation (as shown in figure 4C and 4D). In other words, PARP10 KD slow down 2alpha phosphorylation and SG formation, and thus, the pellets contain less SG material, in particular lower amounts of translation initiation factors. The interpretation of this result should be revised.

Although the isolation of SG using differential centrifugation method (Jain et al, Cell 2016) is challenging, this protocol has been successfully established and utilized by several groups to isolate SGs. Importantly, multiple publications report changes in the SG proteome in different cellular context (Chen et al, NCB 2023; Yang et al, Autophagy 2022; Manivannan et al, JVI 2020; Zhang et al, Cell 2018), and specifically Hu et al. (Nat Comm 2023) demonstrate differential recruitment of SG components over time using this SG isolation protocol. As mentioned in the first round of revision, we will further extend our work to investigate the SG proteome in WT and PARP10 deficient cells in future studies.

We have revised the interpretation in the discussion (line 226) as follows: *"These data suggest that PARP10 modulates SG assembly at multiple levels, ranging from the regulation of eIF2 α phosphorylation to the recruitment of translation factors within SGs."*

PREVIOUS- In summary, the model proposed in Fig4E (as well as the title and abstract) largely overstates the relevance of G3BP phosphorylation in driving SG formation. Instead, the upstream defective eIF2 α phosphorylation may fully explain the effect. If the target residue(s) in G3BP can be identified, a point mutation that abrogates phosphorylation might directly assess the role of G3BP phosphorylation.

RESPONSE -We appreciate the reviewer's thoughtful critique regarding the model presented in Fig. 4F. We agree that reduced eIF2 α phosphorylation upon PARP10 knockdown partly contributes to the observed defect in stress granule formation, and we have now revised the abstract and incorporated these comments in our discussion.

"Regardless, this study emphasizes the crucial role of ADP-ribosylation mediated by PARP10, a MAR-adding enzyme, to maintain the phosphorylation of the translation factor eIF2 α and the presence of other translation factors within the SG core. These data suggest that PARP10 may

mediate SG assembly at multiple steps, from modulation of stress signaling to stress granule condensation and composition."

However, we would like to highlight that our data also indicate that PARP10 regulates SG composition beyond simply modulating translation initiation. Specifically, we show that several translation initiation factors are less efficiently incorporated into SG-enriched fractions despite similar recovery of G3BP1 (Fig. 4E). These findings suggest that PARP10-mediated ADP-ribosylation may influence the biophysical properties of G3BP1 or its ability to recruit interacting partners during SG assembly.

We fully agree that identifying the specific ADP-ribosylation site(s) on G3BP1 and testing their functional relevance via site-directed mutagenesis would provide a more definitive test of this hypothesis. We are actively pursuing this direction, and plan to incorporate these experiments in future studies.

SECOND REVISION -As I said before, the interpretation of the western blots of the pellets enriched in SGs when less SGs are formed is dubious and doesn't support the conclusion. In addition, given the limitations to specifically manipulate of G3BP maryl原因/parylation, while leaving most of the proteome unchanged, authors are not in the position to conclude that G3BP modifications are the cause of the observed effects on SGs.

Please kindly see our response above. Multiple publications have performed WB/proteomic analysis on SG-enriched fractions and identified compositional differences.

The Discussion includes a section entitled "Working Model: MARYlation as a rate-limiting step of SG assembly" dedicated to the maryl原因 of G3BP and its effect on protein-protein interactions and aggregation. The role of 2alpha phosphorylation on SG formation and the effect of PARP10 in this triggering event is surprisingly absent. Once again, I'd like to stress this is an important finding, as PARYaltion is a therapeutic target and knowing the cellular consequences of its manipulation is a valuable contribution.

Likewise, in the 4F figure caption (the model) the importance of G3BP MARYlation doesn't fit with the supporting data, which rather suggest defective 2alpha phosphorylation as the main driver for the slow response observed upon PARP10 KD.

We have now mentioned these points in the discussion section of revised version (line 273-276). As the reviewer suggested, we have revised the statements in the working model section as well:

Line 289 - One scenario is that PARP10 mediates the initial stage of SG assembly through ADP-ribosylation of G3BP1 as well as by regulating eIF2 α phosphorylation.

Line 305 - assembly is initiated by the rate-limiting step of MARYlation—both by modifying substrate proteins and by maintaining p-eIF2 α levels

We have revised the caption in Fig 4F from "PARP10-mediated MARYlation" to "PARP10 initiates SG assembly".

In addition, please modify the following figure titles

Figure 2 | "PARP10 is essential for SG assembly". PARP10 is not essential. The KD just slightly delays SG formation.

Figure 3 | "PARP10-mediated ADP-ribosylation is critical for SG condensation". It's not critical.

We have modified the figure titles as follows:

Figure 2 | "PARP10 knockdown delays SG assembly".

Figure 3 | "PARP10-mediated ADP-ribosylation modulates SG condensation".

PREVIOUS- MINOR:

1). Figure 1D (the effect of MAR-degrading nsP3) and line 98. "a reduction (of PAR signal) in fractions enriched with SG-associated proteins". It appears that all lanes with significant OD (from 2 to 6) have reduced ADPribose signal. (Should we expect an effect specific of SG components?)

RESPONSE-Indeed, there are several SG components that could be ADP-ribosylated as previously reported (PMID: 30100264; PMID: 30728452; PMID: 39760726) and those modification might be reduced by MAR-degrading nsP3. In our current study, we chose to characterize the core SG component G3BP1 (Figs. 3 and 4) since we have previously reported that G3BP1 is ADP-ribosylated during stress (PMID: 21596313) and viral infection (PMID: 33547245). Our future aim is to perform proteomic analysis of ADP-ribosylated SG proteome to define other modified proteins.

SECOND REVISION- The point I made is whether authors propose that the reduced PAR signal (now Fig 1E) was limited to or more dramatic for SG components. Figure 1E doesn't support a selective effect. Rather, as one should expect, all bands in all fractions with significant OD shows reduced PAR signal. This is perhaps another example of the lack of accuracy when interpreting the observations. Authors should reconsider whether a polysome profile is a useful strategy to analyze SG-associated proteins. Please modify line 98 and the overstatement in line 107 accordingly.

Polysome profiling is one of the first methods utilized to isolate SG-associated post translationally modified proteins (Ohn et al, NCB 2008); hence, we employed this approach for our preliminary analysis. This strategy is useful for isolating translationally stalled mRNP-containing complexes (fractions 4 and 5), as described in the main text. However, we agree with the reviewer's concern that PAR signal from non-SG component may also be reduced. As suggested, we have modified lines 98 (now line 108) and 107 (now line 118), specifically adding "SG-associated mRNP complexes" and removed "within the condensates".

September 19, 2025

RE: Life Science Alliance Manuscript #LSA-2024-03026-TRR

Anthony Leung
Johns Hopkins University
615 N. Wolfe Street
BSPH/E8647/Leung Lab
Baltimore, MD 21205

Dear Dr. Leung,

Thank you for submitting your revised manuscript entitled "PARP10 is Critical for Stress Granule Initiation".

We appreciate your constructive engagement with the several concerns of Reviewer 3, primarily related to eIF2a phosphorylation status. We understand that the current work is focused on PARP enzymes and may retain some degree of uncertainty on whether the effects of PARP10 on SGs are related exclusively to its activity towards G3BP1 rather than broader effects on translation. Having considered your responses to this reviewer's concerns, we agree that the several changes to the results and discussion faithfully reflect the evidence for eIF2a involvement without overstating your conclusions. Thank you also for considering inclusion of the puromycilation assays. We understand and accept your hesitation to include these results in your manuscript. In light of these changes, we would be happy to publish your paper in Life Science Alliance pending final revisions necessary to meet our formatting guidelines.

- Please add a Conflict of Interest statement to your main manuscript text.
- Please add a Data Availability section, placed after the Materials & Methods section. For more guidance, consult our guidelines at <https://www.life-science-alliance.org/manuscript-prep#format>
- Please add the X and Bluesky handles of your host institute/organization, as well as your own and/or one of the authors in our system.

A. FINAL FILES:

B. MANUSCRIPT ORGANIZATION AND FORMATTING:

Thank you for your attention to these final processing requirements. Please revise and format the manuscript and upload materials as soon as you are able.

Sincerely,

September 24, 2025

RE: Life Science Alliance Manuscript #LSA-2024-03026-TRRR

Prof. Anthony Leung
Johns Hopkins University
615 N. Wolfe Street
BSPH/E8647/Leung Lab
Baltimore, MD 21205

Dear Dr. Leung,

Thank you for submitting your Research Article entitled "PARP10 is Critical for Stress Granule Initiation". It is a pleasure to let you know that your manuscript is now accepted for publication in Life Science Alliance. Congratulations on this interesting work.

DISTRIBUTION OF MATERIALS:

Again, congratulations on a very nice paper. I hope you found the review process to be constructive and are pleased with how the manuscript was handled editorially. We look forward to future exciting submissions from your lab.

Sincerely,
